# Strong hole-photon coupling in planar Ge for probing charge degree and strongly correlated states

Franco De Palma[1,2,7], Fabian Oppliger[1,2,7], Wonjin Jang [1,2,7], Stefano Bosco [3,4], Marián Janík [5], Stefano Calcaterra [6], Georgios Katsaros [5], Giovanni Isella [6], Daniel Loss [3] & Pasquale Scarlino [1,2] ✉

Semiconductor quantum dots (QDs) in planar germanium (Ge) hetero-structures have emerged as front-runners for future hole-based quantum processors. Here, we present strong coupling between a hole charge qubit, defined in a double quantum dot (DQD) in planar Ge, and microwave photons in a high-impedance ($Z_r = 1.3$ k$\Omega$) resonator based on an array of superconducting quantum interference devices (SQUIDs). Our investigation reveals vacuum-Rabi splittings with coupling strengths up to $g_0/2\pi = 260$ MHz, and a cooperativity of $C \sim 100$, dependent on DQD tuning. Furthermore, utilizing the frequency tunability of our resonator, we explore the quenched energy splitting associated with strong Coulomb correlation effects in Ge QDs. The observed enhanced coherence of the strongly correlated excited state signals the presence of distinct symmetries within related spin functions, serving as a precursor to the strong coupling between photons and spin-charge hybrid qubits in planar Ge. This work paves the way towards coherent quantum connections between remote hole qubits in planar Ge, required to scale up hole-based quantum processors.

Semiconductor quantum dots (QDs) represent a promising platform for advanced quantum information processing[1–3]. Particularly, hole confinement in QDs enables rapid electric spin manipulation due to the large spin-orbit interaction[4–8]. QD-based hole qubit systems have been implemented in various platforms, including fin field-effect transistors (finFETs)[9,10], Ge/Si core/shell nanowires[5,11], and planar Ge/SiGe heterostructures[4,12,13]. Among these, planar Ge stands out due to its exceptional characteristics[6], including high hole mobility ($\mu > 10^6$ cm$^2$ V$^{-1}$ s$^{-1}$ [14]), low charge noise[15], and a low effective mass[16]. Furthermore, nuclear isotope purification can be performed, effectively mitigating magnetic field noise and enhancing the qubit coherence[6]. Building on all these advantages, recent works have

demonstrated coherent single- and two-qubit operations[4,13], scalable multi-qubit array architecture[12,17], and coherent spin shuttling[18] in planar Ge.

In the context of circuit quantum electrodynamics (cQED), the hybridization of microwave photons in superconducting cavities with QD-based qubits holds enormous potential for various applications in quantum technology. These applications include enabling long-range interactions between distant quantum-dot qubits[19–21], achieving rapid and high-fidelity charge and spin state detection[22,23], as well as facilitating analog quantum simulation of open quantum systems[24], and advancing the development of gigahertz photodetectors[25]. However, achieving strong light-matter coupling is a fundamental prerequisite for

[1]Hybrid Quantum Circuit Laboratory, Institute of Physics and Center for Quantum Science and Engineering, École Polytéchnique Fédérale de Lausanne (EPFL), Lausanne 1015, Switzerland. [2]Center for Quantum Science and Engineering, École Polytéchnique Fédérale de Lausanne (EPFL), Lausanne 1015, Switzerland. [3]Department of Physics, University of Basel, Klingelbergstrasse 82, Basel 4056, Switzerland. [4]QuTech, Delft University of Technology, Delft, The Netherlands. [5]Institute of Science and Technology Austria, Am Campus 1, Klosterneuburg 3400, Austria. [6]L-NESS, Physics Department, Politecnico di Milano, via Anzani 42, Como 22100, Italy. [7]These authors contributed equally: Franco De Palma, Fabian Oppliger, Wonjin Jang. ✉e-mail: pasquale.scarlino@epfl.ch

these endeavors. While several previous experiments have successfully demonstrated strong coupling for electrons hosted in Si[26–28], GaAs[29,30], InAs nanowires[31], and for holes in silicon nanowire transistors[32], the strong coupling of holes in planar Ge has remained elusive[33–35].

Previous hybrid cQED experiments primarily focused on resonators interacting with the ground and first excited states of double quantum dot (DQD) charge- or spin two-level systems. In fact, in typical QD structures, additional single-dot orbital states usually lie at energies higher than $100 h \cdot$ GHz, making them inaccessible to microwave resonators[36]. However, low excitation energies can arise from Coulomb interaction-induced renormalization of orbital energies in single QDs, leading to the formation of strongly correlated states (SCSs)[37,38]. When further enhanced by anisotropic QD confinement[39], these states can lead to excitation energies below $10 h \cdot$ GHz that have been observed in GaAs[40,41], Si[42], and carbon nanotube[43,44] QDs and attributed to Wigner molecular (WM) states[38,45–50]. The emergence of SCSs is a general phenomenon, which can take place in QDs defined in any semiconductor platform[41–43]. Such SCSs have profound implications for quantum information processing, offering an encoding for spin-charge hybrid qubits based on exchange interaction[41,47]. If not properly controlled, it can significantly reduce the fidelity of conventional readout schemes in spin qubits[39]. In Ge, it has been also shown that SCSs enable anomalous splittings of spin energy levels without the need for magnetic fields[51]. These findings suggest that low-lying SCSs

could serve as a valuable interface between QD qubits and superconducting circuits in hybrid architectures.

In this study, we establish strong coupling between a microwave photon and a DQD-based hole charge qubit in a planar Ge/SiGe heterostructure, using a high-impedance frequency-tunable resonator based on superconducting quantum interference devices (SQUIDs)[30]. We explore different DQD configurations and achieve a charge-photon vacuum-Rabi splitting (charge decoherence rate) up to $2g_0/2\pi \sim 520$ MHz (down to $\Gamma/2\pi \sim 57$ MHz). We estimate system cooperativity of $C \sim 100$, among the highest reported for QDs charge-resonator hybrid systems to date[29]. Our device geometry facilitates the formation of SCSs in Ge, unveiling a quenched energy spectrum of SCSs in the DQD. Leveraging the frequency tunability of the SQUID array resonator, we perform resonant energy spectroscopy of SCSs in the DQD and extract their energy spectra. By exploring several pairs of adjacent inter-dot configurations, we observe selective coupling to the resonator based on the parity of the DQD hole number and enhanced coherence times for certain excited SCSs, which we attribute to states with a different spin structure[41,52].

## Results

### Architecture for hybrid circuit QED with holes in planar Ge

Figure 1a shows the hybrid superconductor-semiconductor device fabricated on a Ge/SiGe heterostructure[53]. As shown in Fig. 1b, the 16 nm

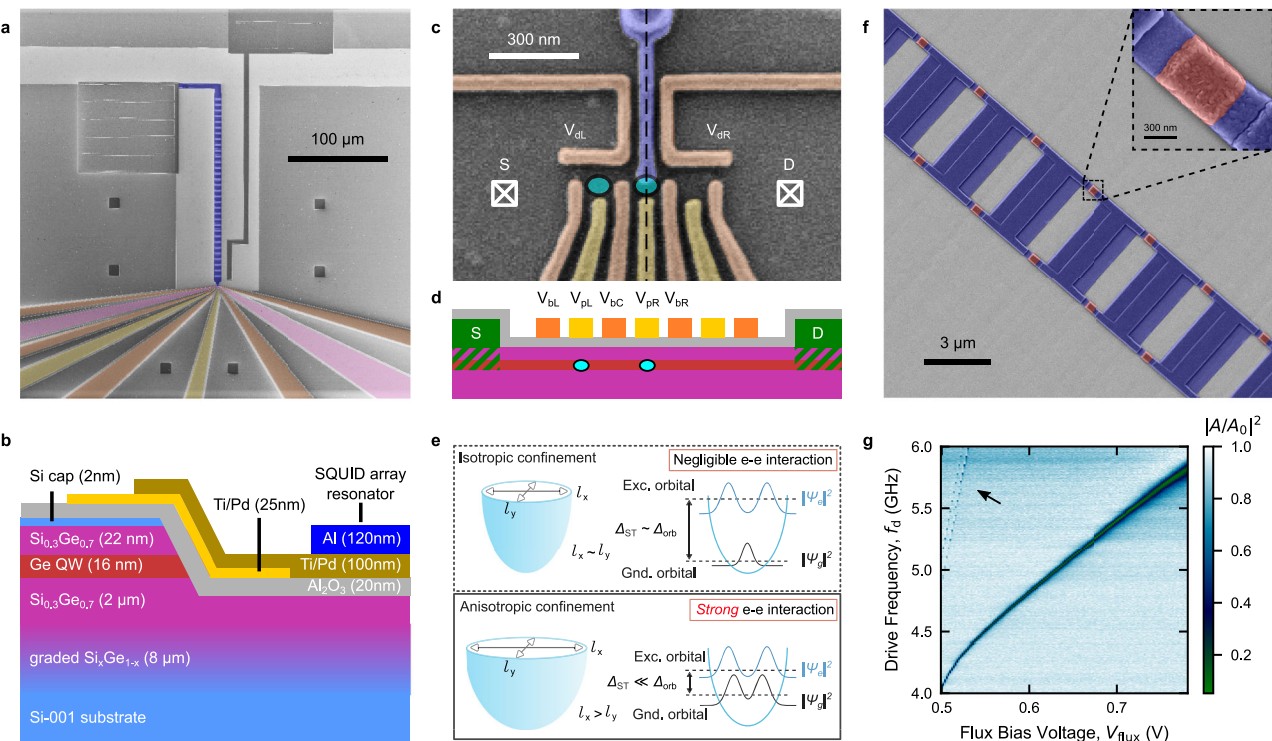

**Fig. 1 | Superconductor-semiconductor hybrid architecture on planar Ge heterostructure. a** False-colored scanning electron micrograph of a representative hybrid device. The SQUID array resonator (violet) is capacitively coupled to the transmission line on top. The QDs are defined electrostatically by barrier (orange) and plunger (yellow) gates. The Ge quantum well is etched away everywhere except for a small mesa region (pink) used to host the QDs. Ohmic contacts are patterned on the extensions of the mesa region. **b** Schematic side-view of the heterostructure and the device across the black dashed line in **c**. **c** False-colored scanning electron micrograph of the QDs region. The expected position of the DQD is highlighted by cyan ellipses. The plunger gate $V_{pL}$ ($V_{pR}$) mainly controls the electrochemical potential of the left (right) QD, while $V_{bL}$ ($V_{bR}$) modulates the tunnel coupling strength of the left (right) QD to left (right) reservoir. $V_{bC}$ controls the inter-dot tunnel coupling strength $t_c$. **d** Side-view of the device across the QD array.

**e** Schematic of the two-body ground and excited state wavefunctions ($\psi_g$ and $\psi_e$) and single QD energy splitting for two different classes of QD confinement potential. Under isotropic confinement, the ground and excited state wavefunctions have distinct shapes, which result in large orbital splitting $\Delta_{orb}$. In the anisotropic and strongly interacting case, the symmetry of $\psi_g$ is broken, resulting in a quenched singlet-triplet splitting $\Delta_{ST} \ll \Delta_{orb}$[39]. **f** False-colored scanning electron micrograph of the SQUID array resonator. Inset: Zoom-in of a single Josephson junction (red). **g** Flux tunability of the SQUID array resonator. Normalized amplitude of feedline transmission $|A/A_0|^2$ as a function of drive frequency $f_d$ and bias voltage $V_{flux}$ applied to the superconducting coil mounted perpendicularly to the sample (see Supplementary Note 1). Higher resonator modes are visible near the half-flux point (black arrow)[66]. The device is operated in a dilution refrigerator with a base temperature of 10 mK (see Supplementary Note 1).

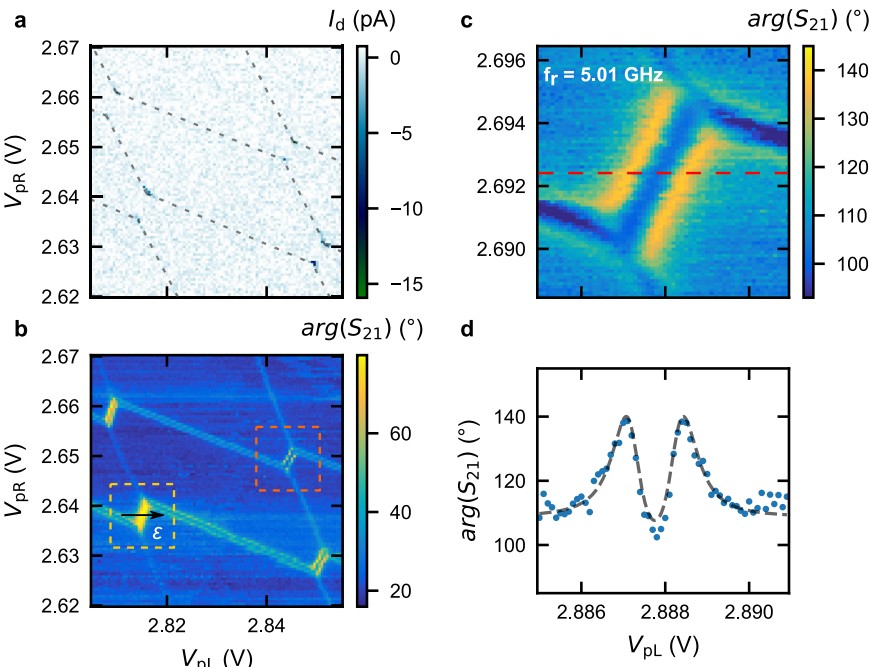

**Fig. 2 | DQD characterization with the tunable resonator.** A region of the DQD charge stability diagram as a function of the applied plunger gate voltages $V_{pR}$ and $V_{pL}$, recorded by dc-transport (**a**) and by measuring the phase (**b**) of the feedline transmission $S_{21}$, $arg(S_{21})$, at $f_d = f_r = 5.01$ GHz. The resonator detects inter-dot and reservoir-dot transitions when their tunneling rates are close to $f_r$[67]. Yellow (orange) dashed box in **b**: the phase signal increases (decreases) near the inter-dot region

with respect to the background, if the resonator is dispersively shifted to lower (higher) frequency. Notably, because the resonator gate lever arm is larger for the right QD, the resonator is more sensitive to its QD-reservoir transitions with respect to those of the left QD. **c**, Inter-dot transition probed with $f_r = 5.01$ GHz - $2t_c/h$. **d** A line-cut taken along the red dashed line in **c**. The black dashed curve shows the fit to a master equation (see Methods).

Ge quantum well (QW), hosting the 2-dimensional hole gas (2DHG), is ~24 nm below the surface. A conductive channel, defined by selectively etching the Ge QW, hosts a DQD (cyan ellipses in Fig. 1c, d) defined by metallic gate electrodes. The gate layout of our device supports relatively large QDs (radius $l_{QD}$ ~ 70 nm). The Wigner ratio $\lambda_W = E_{ee}/E_{orb} \propto l_{QD}$ (see Supplementary Note 6) quantifies the ratio between the Coulomb interaction strength ($E_{ee} \propto 1/l_{QD}$) and the orbital confinement energy ($E_{orb} \propto 1/l_{QD}^2$). Coulomb interactions become increasingly relevant in large QDs, as the ones studied here. In our experiment, we estimate $\lambda_W$ ~ 4.46. Coulomb correlation renormalizes the energy of orbital states in QDs, thus quenching the orbital splitting and, therefore, the singlet-triplet splitting $\Delta_{ST}$[37]. Furthermore, anisotropic QD confinement is expected to enhance the correlation effect and reduce $\Delta_{ST}$ even further (see Supplementary Note 6)[39], as illustrated in Fig. 1e. Orbital state renormalization induced by Coulomb correlation and confinement anisotropy is expected to significantly alter also the charge density distribution of the ground state, promoting the formation of Wigner molecular (WM) states (see Supplementary Fig. 11)[39,40,42–44].

The right dot is coupled to the superconducting resonator (Fig. 1f) via the violet electrode in Fig. 1c (see Supplementary Note 1)[30]. This is designed to maximize the capacitive coupling by completely overlapping one QD and, therefore, to efficiently couple to the DQD via transverse charge-photon interaction[26,29,30]. The resonator consists of an array of $N = 32$ SQUIDs (Fig. 1f) with an inductance of $L$ ~ 0.63 nH/SQUID, resulting in an equivalent lumped impedance of $Z_r$ ~ 1.3 kΩ[30]. The high-impedance resonator enhances the charge-photon coupling strength $g_0$ by maximizing the vacuum voltage fluctuation $V_{0,rms} = 2\pi f_r \sqrt{\hbar Z_r/2}$, according to the relation $g_0 = \frac{1}{2}\beta_r V_{0,rms}/\hbar$, with $\beta_r$ the resonator differential lever arm[32]. The resonator is also capacitively coupled to a 50 Ω waveguide (the photon feedline) on one side, and grounded on the other end, forming a hanged quarter-wave resonator (Fig. 1a)[30]. We probe the microwave response of the hybrid system, recording the feedline transmission ($S_{21}$) at powers

corresponding to less than one photon on average in the resonator (see Supplementary Note 3). By leveraging the external magnetic flux dependence of the critical current of the SQUIDs[54], the resonator frequency $f_r$ can be tuned from ~6 GHz to well below 4 GHz (see Fig. 1g). To apply a finite magnetic flux, we place a superconducting coil on top of the device which generates an out-of-plane magnetic field of 50 ~ 70 µT (see Supplementary Note 1).

Figure 2a shows a region of the DQD stability diagram spanned by $V_{pR}$ and $V_{pL}$, measured by probing the dc current through the DQD[55]. To characterize the charge-photon coupling, we simultaneously monitor the feedline transmission at the frequency $f_d = f_r = 5.01$ GHz (see Fig. 2b). While the dc-transport measurement for the explored configuration only exhibits the DQD triple points[55], the resonator response reveals not only the inter-dot transitions but also the QD-reservoir ones, facilitating an extensive characterization of QD devices. Supplementary Fig. 3 reports a zoom-out of the charge stability diagram shown in Fig. 2a, b.

Close to an inter-dot transition, the DQD system can be approximated by a simplified $2 \times 2$ charge qubit Hamiltonian, given by $H_{cq} = \frac{\varepsilon}{2}\sigma_z + t_c\sigma_x$ with corresponding eigenenergies $E_\pm = \pm\frac{1}{2}\sqrt{\varepsilon^2 + 4t_c^2}$. Here, $\sigma_i$ represents the Pauli operator ($i = x, y, z$)[30] and $\varepsilon$ ($t_c$) is the DQD energy detuning (tunnel coupling). The transverse charge-photon interaction $H_{int} = \hbar g_{eff}(a^\dagger\sigma^- + a\sigma^+)$, with $g_{eff} = 2g_0 t_c/(E_+ - E_-)$ denoting the effective charge-photon coupling strength, hybridizes the qubit with the resonator (see Supplementary Note 4). As a result, the phase of the feedline transmission $S_{21}$ (Fig. 2b) exhibits a different response depending on whether the qubit energy is higher (yellow dashed box) or lower (orange dashed box) than the bare resonator energy.

While tuning the qubit frequency, $f_q = (E_+ - E_-)/h = \sqrt{\varepsilon^2 + 4t_c^2}/h$, to be close to $f_r$ ($|f_q - f_r| < 10g_0/2\pi$) is essential to ensure a significant dispersive resonator response, it can be challenging to achieve depending on the DQD gate layout[32]. The tunable resonator presented

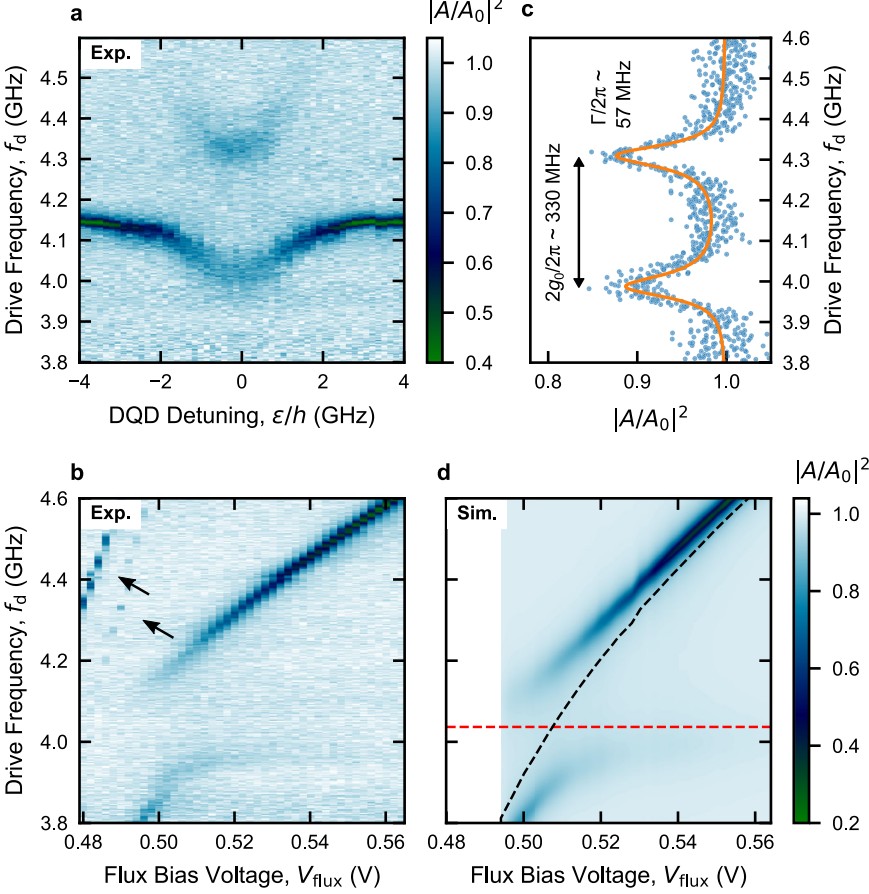

**Fig. 3 | Strong charge-photon coupling at the charge sweet spot. a** Normalized amplitude of feedline transmission $|A/A_0|^2$ as a function of drive frequency $f_d$ and DQD detuning $\varepsilon$. An avoided crossing - the signature of the strong coupling regime–is observed when the DQD-charge transition matches the bare resonator frequency $f_r$. **b** $|A/A_0|^2$ as a function of drive frequency $f_d$ and the voltage $V_{flux}$ applied to the resonator coil, which tunes the resonator frequency $f_r$. During the measurement, the DQD is kept at $\varepsilon = 0$. An avoided crossing is observed around $V_{flux} = 504$ mV, when the bare resonator frequency $f_r$ matches the DQD charge

transition ($f_r = f_q = 2t_c/h$). Higher resonator modes are visible near the half-flux point (black arrows)[66]. **c** $|A/A_0|^2$ as a function of $f_d$ at the resonance condition, highlighting the vacuum-Rabi splitting $2g_0/2\pi$. A fit to the master equation model is represented by a solid orange line (see Methods). All the extracted values are reported in Supplementary Table 1. $2g_0$ and $\Gamma$ are indicated ($2t_c/h = 4.149$ GHz). **d** Simulation of $|A/A_0|^2$ using input-output theory with the parameters $g_0/2\pi = 154$ MHz, $\Gamma/2\pi = 80$ MHz, $t_c/h = 2.018$ GHz extracted from fitting a line-cut of **b** at $V_{flux} = 504$ mV (reported in Supplementary Fig. 8b).

here offers an additional means to efficiently investigate qubits by varying $f_r$ across $f_q$. In Fig. 2c, we record the phase of the feedline transmission, $\arg(S_{21})$, taken with a resonator frequency of $f_r = 5.01$ GHz - $2t_c/h$ to reconstruct the DQD stability diagram of an inter-dot transition. The corresponding line-cut along the red dashed line is reported in Fig. 2d. Leveraging the frequency tunability of our resonator, we also measure the same region of the DQD stability diagram with $f_r$ tuned above and below $2t_c/h$, reported in Supplementary Fig. 4. Line-cuts across the inter-dot transition (Fig. 2d and Supplementary Fig. 4g–i) are simultaneously fitted to a master equation model (denoted by black dashed lines), extracting a common tunnel coupling of $t_c/h = 2.472$ GHz, qubit decoherence $\Gamma/2\pi = 120$ MHz, and charge-photon coupling strength $g_0/2\pi = 192$ MHz for $f_r = 5.01$ GHz - $2t_c/h$ (see Methods). To quantify the quality of the DQD-resonator interface, we evaluate the ratio between the coupling and the decoherence rates, by computing the cooperativity $C = 4g_0^2/(\kappa\Gamma)$[56]. Using $\kappa/2\pi = 30$ MHz, extracted from a bare resonator fit at 5 GHz, along with the aforementioned parameters, we estimate $C \sim 40 \gg 1$, indicating the possibility of observing strong coupling.

## Strong hole charge-photon coupling

We now probe the charge-photon interaction at $\varepsilon = 0$ (charge sweet spot), where the electric dipole moment of the holes in the DQD is

maximal, resulting in a vacuum-Rabi mode splitting of $2g_{eff} = 2g_0$[26,30]. Figure 3a shows the normalized feedline transmission amplitude $|A/A_0|^2$ as a function of $f_d$ and with $\varepsilon$ changed to cross an inter-dot transition (as depicted by the black arrow in Fig. 2b). We note that, as we detail in Fig. 4 below, the two subsystems are not perfectly in resonance at $\varepsilon = 0$ in Fig. 3a. Our resonator's frequency tunability offers a convenient way to investigate vacuum-Rabi splitting while keeping the DQD electrostatic configuration constant. This allows us to reach the resonant condition between the DQD two-level system and the resonator, while keeping the DQD gate voltages unchanged. Thereby, we fix the detuning at $\varepsilon = 0$ and vary the external magnetic flux to fine-tune the resonator frequency $f_r$ into resonance with the qubit frequency $f_q = 2t_c/h$. In Fig. 3b, we report $|A/A_0|^2$ as a function of $f_d$ and flux bias voltage $V_{flux}$, where the charge-photon hybridization at ∼4 GHz results in a clear vacuum-Rabi mode splitting. By fitting a line-cut of Fig. 3b taken at $V_{flux} = 504$ mV (reported in Supplementary Fig. 8b), we extract the parameters $(t_c/h, g_0/2\pi, \Gamma/2\pi) = (2018, 154, 80)$ MHz. These parameters are utilized to numerically reconstruct $|A/A_0|^2$ (Fig. 3d). To better evaluate the cooperativity of our system, in Fig. 3c we report a high-quality vacuum-Rabi mode splitting measured, with increased averaging, as a function of $f_d$ in the same DQD configuration ($\varepsilon = 0$), but at $V_{flux} = 507$ mV to compensate for a slight drift in qubit frequency. By fitting to the master equation model (solid line in Fig. 3c, see Methods),

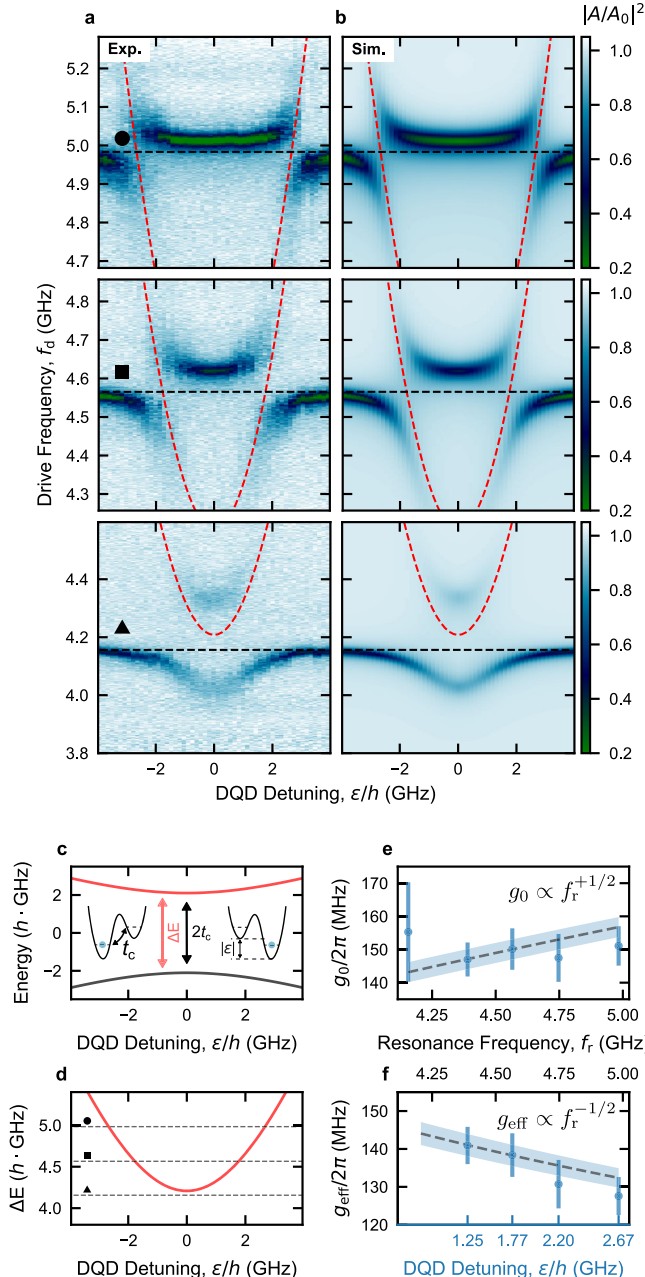

**Fig. 4 | Charge qubit spectroscopy via tunable resonator. a** Normalized amplitude of feedline transmission $|A/A_0|^2$ as a function of drive frequency $f_d$ and DQD detuning $\varepsilon$. The three panels are taken in correspondence of three different bare resonator frequencies $f_r$, denoted by black circle, square and triangle, while keeping the inter-dot tunnel coupling $t_c$ constant. **b** Simulation of $|A/A_0|^2$ using input–output theory with the parameters extracted by fitting the full dataset in the corresponding panels in **a** to the master equation model (see Methods). Black (red) dashed lines in **a**, **b** denote the bare resonator (DQD-charge qubit) frequency. **c** Energy-level diagram, i.e., the energy spectrum, of the DQD charge qubit system as a function of DQD detuning $\varepsilon$ (calculated for the charge qubit Hamiltonian in main text). The black (red) curve represents the ground (excited) state of the charge qubit. Inset: DQD potential schematics showing the charge state at the negative and positive $\varepsilon$. **d** Excitation energy $\Delta E$ as a function of DQD detuning $\varepsilon$. The dashed lines denoted by black circle, square and triangle correspond to different $f_r$ in **a**. **e** Extracted charge-photon coupling strength $g_0$ as a function of $f_r$. **f** Effective charge-photon coupling strength $g_{\text{eff}}$ at the $\varepsilon$ values for which the two subsystems are in resonance, estimated using $g_{\text{eff}} = g_0 t_c / h f_r$. The DQD detuning values corresponding to the avoided crossings are indicated at the bottom axis in blue. Since the resonance condition is not met for the lowest panel in **a** ($f_r < 2t_c/h$), the first point in **f** is omitted. The dashed lines in **e**, **f** represent the expected trend of the coupling strengths as a function of $f_r$, while the error bars and the shaded regions indicate the estimated uncertainties (see Methods).

indicating a higher coupling of the resonator to the detuning degree of freedom in the second case, albeit at the cost of a larger $\Gamma$[29].

**Tunable high-impedance resonator for qubit spectroscopy**

We leverage the resonator frequency tunability to conduct resonant energy spectroscopy of the DQD charge qubit, in the same DQD configuration as in Fig. 3, and keeping the DQD at a fixed $t_c$[30]. This spectroscopy aims to reconstruct the qubit's energy dispersion. In contrast to the measurements in Fig. 3, where $f_r \sim 2t_c/h$, here we extend our investigation also to higher resonator frequencies, $f_r > 2t_c/h$.

In Fig. 4a, we present the measured normalized feedline transmission $|A/A_0|^2$ as a function of $f_d$ and $\varepsilon$ for three different values of $f_r$, as indicated by dashed lines in Fig. 4d. The schematics in Fig. 4c, d illustrate the charge qubit energy-level diagram (panel c) and the excitation energy spectrum $\Delta E = E_+ - E_- = E_{cq}$ (panel d) along $\varepsilon$. Notably, clear avoided crossings are observed in Fig. 4a when the charge qubit gets in resonance with the resonator ($hf_r = \sqrt{\varepsilon^2 + 4t_c^2}$). Additional details on the charge qubit spectroscopy are available in Supplementary Note 5.

These spectroscopy measurements also provide valuable insights into the evolution of the effective charge-photon coupling strength $g_{\text{eff}}$, as a function of the DQD detuning $\varepsilon$. By fitting all five datasets presented in Supplementary Fig. 10a to the master equation model, we accurately reproduce the hybridized charge qubit-resonator spectra, as shown in Fig. 4b and Supplementary Fig. 10b. For this fit, the full 2D datasets are considered and a detuning dependence of the qubit decoherence rate $\Gamma(\varepsilon)$ is included in the model (see Methods for more details).

From these spectra, we extract the charge-photon coupling strengths $g_0$, and present them as a function of $f_r$ in Fig. 4e. We also estimate the effective charge-photon coupling strengths $g_{\text{eff}} = g_0 2t_c / \sqrt{\varepsilon^2 + 4t_c^2} = g_0 2t_c / hf_r$, when the two systems are in resonance, and report them as a function of both $\varepsilon$ and $f_r$ in Fig. 4f. Since $g_0 = \frac{1}{2}\beta_r V_{0,\text{rms}}/\hbar = \frac{1}{2}\beta_r 2\pi f_r \sqrt{Z_r/(2\hbar)}$, where the lumped equivalent resonator impedance $Z_r$ can be written in terms of $f_r$ as $Z_r = 1/(2\pi f_r C_r)$, we obtain a frequency dependence of $g_0 \propto \sqrt{f_r}$ and hence $g_{\text{eff}} \propto 1/\sqrt{f_r}$ (assuming constant $C_r$ and resonance condition). The dashed line in Fig. 4e represents a fit of the extracted $g_0$ to the expected frequency dependence. Using this fit, we estimate the evolution of $g_{\text{eff}}$ as a function of $f_r$ and illustrate it as a dashed line in Fig. 4f. Notably, the

we find $(t_c/h, g_0/2\pi, \Gamma/2\pi) = (2072, 165, 57)$ MHz. These parameters result in the cooperativity of $C \sim 100$ (with $\kappa/2\pi = 19$ MHz), which highlights the strong charge-photon coupling in planar Ge.

In Supplementary Fig. 9a, we explore an alternative DQD charge transition, which features an enhanced $g_0$. Fitting the line-cut in Supplementary Fig. 9b to the master equation model, we extract the parameters $(t_c/h, g_0/2\pi, \Gamma/2\pi) = (2711, 260, 192)$ MHz, and calculate a cooperativity of $C \sim 23$ (with $\kappa/2\pi = 63$ MHz). Here, the high $g_0/2\pi = 260$ MHz, enabled by the high-impedance SQUID array resonator, allows us to achieve a strong coupling regime, in spite of a substantial qubit decoherence rate $\Gamma$. We speculate that the difference between the values of $g_0$ and $\Gamma$ extracted from the two datasets in Fig. 3 and Supplementary Fig. 9 may arise from distinct effective electric dipole moments associated with the two DQD electrostatic configurations[29]. To account for the frequency dependence of the coupling strength between the resonator and DQD, we calculate the resonator's differential lever arm $\beta_r = \frac{2g_0 h}{V_{0,\text{rms}}}$ in the two configurations. We find $\beta_r$ values of 0.18 and 0.25 eV/V (see Methods), respectively,

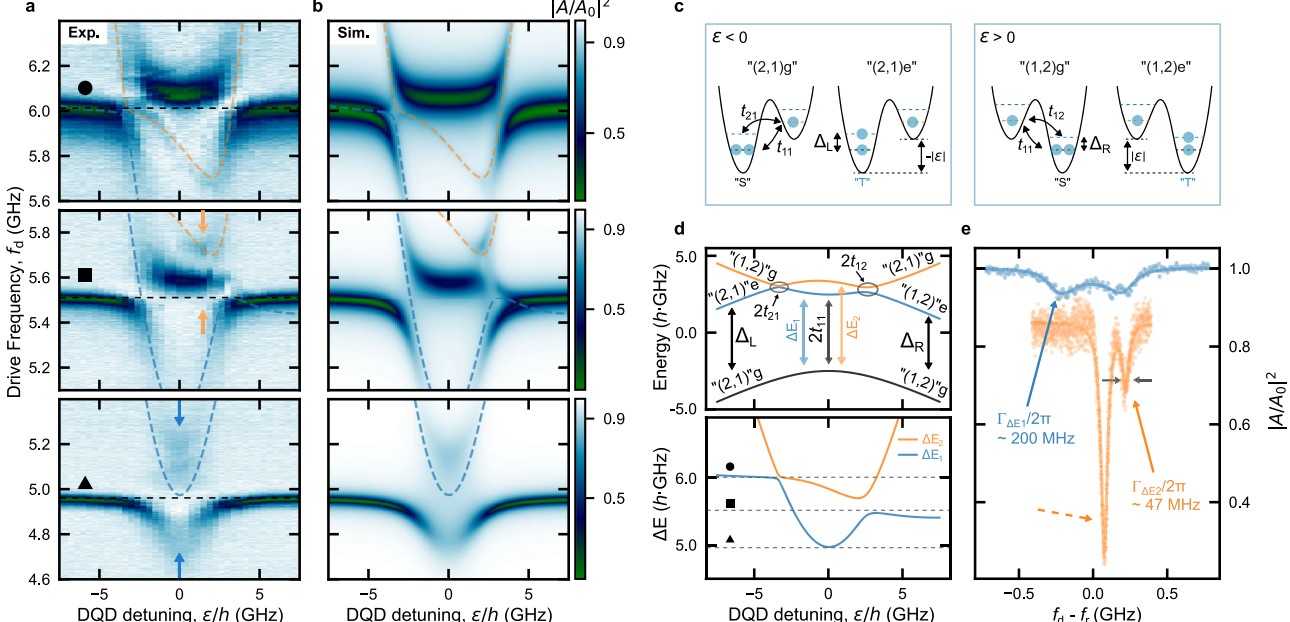

**Fig. 5 | Spectroscopy of the strongly correlated states in the hybrid architecture. a** Normalized amplitude of feedline transmission $|A/A_0|^2$ as a function of drive frequency $f_d$ and DQD detuning $\varepsilon$. The three panels are taken in correspondence of three different bare resonator frequencies $f_r$ (black dashed lines denoted by black circle, square and triangle). The dashed blue and orange lines show the calculated excitation spectra of the DQD, as detailed in **d**, revealing the presence of quenched strongly correlated states (SCSs). **b** Simulated $|A/A_0|^2$ using a generalized input-output theory of a multi-level DQD system (see Methods, and Supplementary Note 7) and for the three different $f_r$ as in **a**. The parameters for the simulations can be found in Supplementary Table 3. **c** DQD schematics of the states relevant to "(2, 1)" and "(1, 2)" charge configurations for negative and positive DQD detuning $\varepsilon$. $\Delta_L$ ($\Delta_R$) is the singlet-triplet energy splitting $\Delta_{ST}$ when two holes are paired in the left

(right) QD. **d** Energy-level diagram (top panel) and excitation energy $\Delta E$ (bottom panel) calculated with the $4 \times 4$ Hamiltonian in Methods, and used for the input-output simulation in **b**. In the bottom panel, the blue (orange) curve corresponds to the energy splitting $\Delta E_1$ ($\Delta E_2$) between the first (second) excited state branch and the ground state, shown in the top panel. The black dashed lines denoted by a black circle, square, and triangle in the bottom panel represent the different $f_r$ used for acquiring the distinct spectra in **a**. **e** Frequency line-cut taken at the DQD detuning indicated by the blue (orange) arrows in **a**. The blue (orange) data highlights the resonator hybridization with the $\Delta E_1$ ($\Delta E_2$) transition. A fit to the master equation model (solid blue line), and Lorentzian (solid orange line) results in $\Gamma_{\Delta E_1}/2\pi \sim 200$ MHz and $\Gamma_{\Delta E_2}/2\pi \sim 47$ MHz, respectively. The orange dashed arrow indicates the resonator dispersively shifted by the interaction with the charge-like excitation $\Delta E_1$.

evolution of $g_0$ does not closely follow the expected trend as a function of $f_r$. This discrepancy can be attributed to a nonuniform and frequency-dependent voltage profile of the resonator mode, potentially due to magnetic flux inhomogeneity along the SQUID array. Alternatively, simultaneous hybridization of the DQD with higher order resonator modes (see Fig. 1g), which approach the qubit frequency in the studied flux range, may influence the coupling to the fundamental mode. Further investigation is required in order to better understand the evolution of $g_0$.

**Hybrid circuit QED with SCSs**

Strikingly, our investigation of multiple adjacent inter-dot transitions reveals that the conventional charge qubit-like spectroscopy, as illustrated in Fig. 4, featuring a single two-level system coupled to the resonator, fails to describe several cases. For instance, in Fig. 5a, we present three independent measurements of the normalized feedline transmission $|A/A_0|^2$ as a function of $f_d$ and $\varepsilon$, obtained for the same DQD configuration, but in correspondence to three different resonance frequencies $f_r$ (indicated by black dashed lines denoted by a black circle, square, and triangle). See Supplementary Fig. 12 for a more detailed resonator spectroscopy. These measurements unveil unconventional features, including anomalous spectroscopy diagrams asymmetric in $\varepsilon$, additional avoided crossings, and distinct spectroscopic lines that deviate significantly from the conventional model for a resonator hybridized with a two-level system and have not been previously documented.

The anomalous spectrum of these DQD configurations is captured by an extended model that includes an excited state in each QD, whose energies can be close to $hf_r$. Specifically, we adopt a $4 \times 4$ Hamiltonian

similar to the one used in prior studies[52,57] and numerically simulate the DQD spectrum and feedline transmission. We assume the validity of an effective hole numbering, where we neglect the even core holes (see Methods). More specifically, in modeling Fig. 5a, we assume a "(2,1)"↔"(1,2)" DQD charge configuration (odd case). Within the "(2,1)" configuration, the two holes in the left QD can occupy either the ground orbital state, forming "(2,1)g", or the excited orbital state, forming "(2,1)e" (see Fig. 5c, for $\varepsilon < 0$). Analogously, for $\varepsilon > 0$, the eigenstates consist of the ground and excited states of the right QD, corresponding to "(1,2)g" and "(1,2)e", respectively (see Fig. 5c, for $\varepsilon > 0$).

As we demonstrate in detail below, making use of Fig. 6, such an effective particle numbering in the QDs readily captures spin structures that depend on the hole number parity[32,53,58]. In this regard, we expect distinct spin symmetries in our "(2,1)"↔"(1,2)" configuration related to the ground and excited states[52]. More explicitly, we assume the ground (excited) state to involve anti-symmetric singlet "S" (symmetric triplet "T") spin pairing in the doubly-occupied QD (Fig. 5c). In this configuration, the two lowest energy levels form doublet spin states together with the single spin in the other QD[52]. For instance, "(2,1)g" ("(2,1)e") forms a doublet state with spin singlet (triplet) pairing in the left QD. Here, finite exchange interaction can couple the ground and excited doublet states[52,57], because they have the same total spin quantum number $S_{tot} = 1/2$, despite the different spin symmetries within the doubly-occupied QD. With this exchange interaction, the "(2,1)g"↔"(2,1)e" or "(1,2)g"↔"(1,2)e" transitions can be revealed by our resonator as presented in Fig. 5a, in agreement with the spin selection rule.

Based on the above modeling, we empirically determine the Hamiltonian parameters, including $\Delta_L/h = 5.40$ GHz ($\Delta_R/h = 4.73$ GHz),

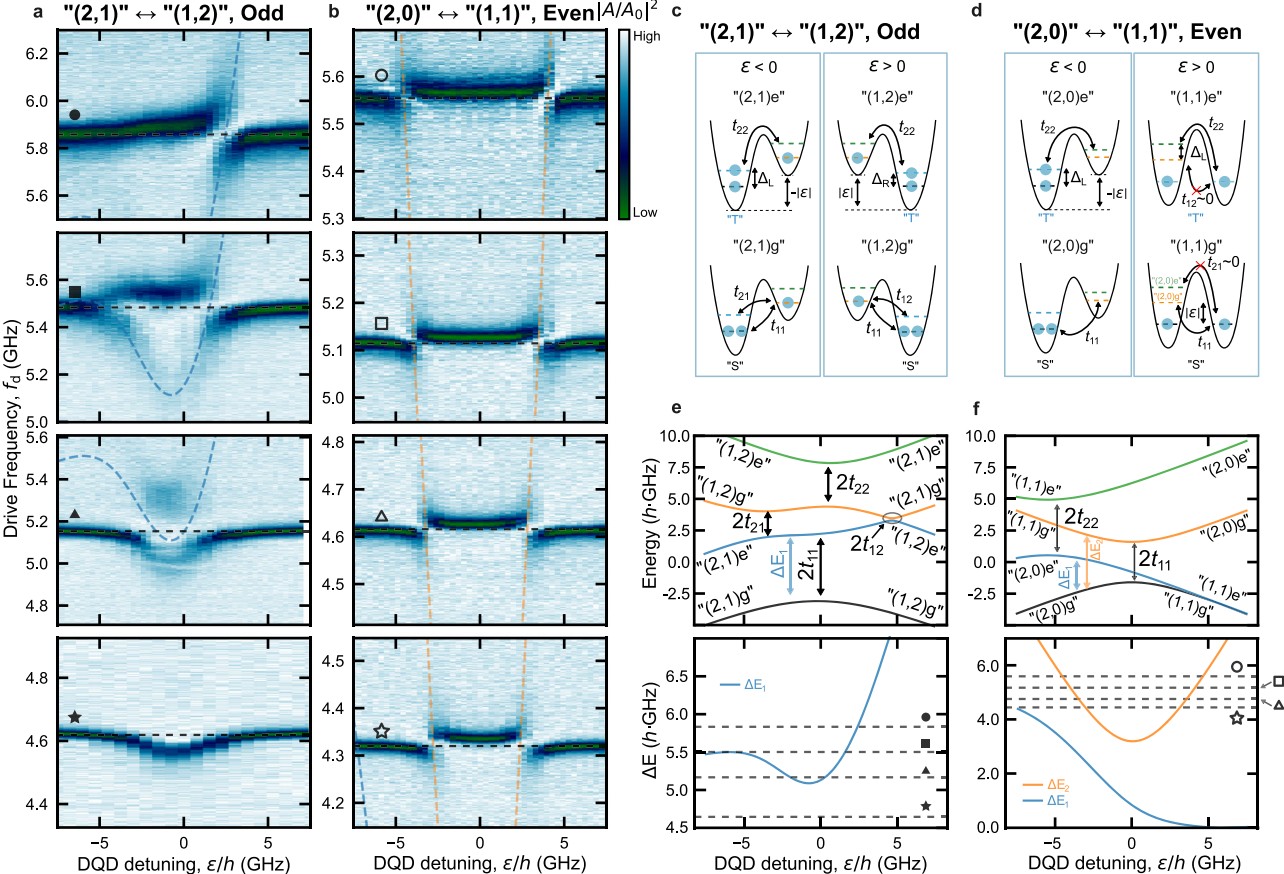

**Fig. 6 | Hole number parity-dependent behavior. a, b** DQD resonant spectra measured for two adjacent inter-dot transitions "(2, 1)"↔"(1, 2)" (odd, **a**) and "(2,0)"↔"(1,1)" (even, **b**). Each panel presents the normalized amplitude of feedline transmission $|A/A_0|^2$ as a function of drive frequency $f_d$ and DQD detuning $\varepsilon$, obtained in correspondence with the bare resonator frequency indicated by the horizontal black dashed line. The dashed blue and orange lines show the calculated first, and second excitation spectra of the DQD, as detailed in **e, f**, revealing the presence of quenched SCSs. **c, d** Schematics of the relevant states for "(2, 1)"↔"(1, 2)" (**c**, identical to the one shown in Fig. 5c) and "(2,0)"↔"(1,1)" (**d**) inter-dot transition. In the even parity case **d**, the energy gap between "(1,1)g" and "(1,1)e", $\Delta_R \sim 0$ due to the negligible exchange interaction of the unpaired holes. $\Delta_L$ is fixed to

the value used in the odd case because the number of the holes in the left QD is unchanged. In the "(1, 1)" charge state, "S" ("T") denotes the spin singlet (triplet) state formed by the two holes in the respective QD. $t_{12}, t_{21} \sim 0$ due to the spin selection rule. **e, f** Energy-level diagram (top panel) and excitation energy $\Delta E$ (bottom panel) calculated with the $4 \times 4$ Hamiltonian described in Methods, to obtain the resonant spectra in **a, b**, respectively. In the bottom panel, the blue (orange) curve corresponds to the energy splitting $\Delta E_1$ ($\Delta E_2$) between the first (second) excited state branch and the ground state, shown in the top panel. The black dashed lines denoted by the different black-filled (blank) symbols in the bottom panel of **e** (**f**) represent the different $f_r$ used for acquiring the distinct spectra in **a** (**b**).

i.e. the singlet-triplet splitting $\Delta_{ST}$ in the left (right) QD, that accurately reproduce both the energy and excitation spectra, reported respectively in the top and bottom panels of Fig. 5d (see Supplementary Table 4). We also estimate the tunnel coupling strengths between the $i$th state of the left QD and the $j$th state of the right QD, $t_{ij}$. Furthermore, we use input-output theory to analyze the interaction between the resonator and the multi-level QD system[59] (see Methods and Supplementary Note 7), enabling us to accurately reproduce the spectrum of the hybridized system, as depicted in Fig. 5b. Note that for this, it is essential to assume finite tunnel coupling strengths $t_{12}/h = 0.21$ GHz and $t_{21}/h = 0.11$ GHz.

The extracted values of $\Delta_{L,R}$ are orders of magnitude smaller than the expected orbital energy gap (~70$h \cdot$GHz) obtained from a single particle model, considering the dimensions of our QDs (see Supplementary Note 6). Instead, these estimated excitation energies are generated by strong Coulomb correlation effects within each QD. To support this interpretation, in Supplementary Note 6, we present a preliminary model for two interacting holes in planar Ge, which suggests that the small anisotropy in the QD confinement, in conjunction with electron-electron interactions, can result in SCSs with $\Delta_{ST} \lesssim 10h \cdot$GHz. Although a more comprehensive investigation based

on full-configuration-interaction calculations is necessary to precisely characterize the energy scales within the DQD[42,47], our preliminary analysis provides evidence that the observed features in Fig. 5a may be attributed to WM states[38,45–50].

As described above, we note that SCSs exhibit distinct spin symmetries, with the ground orbital state supporting the anti-symmetric spin singlet, and the excited orbital state supporting the symmetric spin triplet[40,52] (see Fig. 5c). These symmetries imply that the relaxation between the states specified above involves a spin-changing process, which can be considerably slower compared to the bare charge relaxation[40,52]. We explore this distinction in Fig. 5e, which presents two line cuts along $\varepsilon$, marked by the orange and blue dashed lines in the middle and bottom panels of Fig. 5a. In comparison to the charge qubit-like decoherence rate $\Gamma_{\Delta E_1}/2\pi \sim 200$ MHz extracted from the fit to the master equation (blue solid line in Fig. 5e), the second excited state spectrum $\Delta E_2$ (denoted by an orange solid arrow in Fig. 5e) is characterized by a significantly narrower linewidth $\Gamma_{\Delta E_2}/2\pi \sim 47$ MHz (extracted from the fit to the Lorentzian), which is further supporting our modeling. Similar spectroscopic signatures, attributed to the SCSs with excitation energies very close to that of the resonator, have been detected across multiple inter-dot transitions. Supplementary Fig. 13

reports another instance of a similar spectrum exhibiting $\Gamma_{\Delta E_2}/2\pi \sim 35$ MHz.

The presented Hamiltonian also models a spin-charge hybrid qubit, which can be encoded in the SCSs exhibiting a lower decoherence rate in comparison to a bare charge qubit. Such a hybrid qubit also allows all-electrical control of the spin states based on exchange interaction[41,52].

To further explore these unconventional DQD spectra due to strong Coulomb correlation effects, and to confirm the intrinsic spin nature of the aforementioned states, we delve into the expected hole number parity-dependence, distinguishing between even and odd effective DQD occupation. As we detail below, the observed energy spectra measured with our resonator, both in the even and odd configurations, are consistent with the spectra derived from our effective model, which takes into account the parity-dependent spin structures. In Fig. 6a and b, we investigate a representative instance of two neighboring inter-dot transitions involving effective charge configurations "(2, 1)"↔"(1, 2)" and "(2, 0)"↔"(1, 1)", respectively (denoted in the stability diagram in Supplementary Fig. 14a by the dashed black and red boxes).

In the "(2, 1)"↔"(1, 2)" configuration (Fig. 6a), characterized by an odd total number of holes, the ground and first excited states have the same spin quantum number $S_{tot} = 1/2$. Similar to the configuration shown in Fig. 5, this results in a finite exchange interaction between "(2,1)g" and "(2,1)e", and between "(1,2)g" and "(1,2)e", enabling their electrical coupling to the resonator, in accordance with the spin selection rule. To faithfully replicate both energy and excitation spectra shown in Fig. 6e, it is essential to assume a sizable $t_{21}$ and a relatively small $\Delta_L/h \sim 5.48$ GHz. In contrast to Fig. 5a, the resonator spectroscopy reported in Fig. 6a does not show the second excitation $\Delta E_2$, due to the larger $\Delta_R$ with respect to $f_r$.

For the adjacent even configuration, denoted as "(2, 0)"↔"(1, 1)" (see Fig. 6b, d), with a single additional hole in the right QD compared to the odd configuration, the total spin numbers of the ground ($S_{tot} = 0$) and first excited states ($S_{tot} = 1$) in the DQD are different. This is further supported by the observed signatures of Pauli spin blockade (PSB) presented in Supplementary Note 9. Because the number of holes in the left QD is the same as in the configuration presented in Fig. 6a, we expect a similar $\Delta_L$ between the "(2,0)g" and "(2,0)e" states. The extracted width of the PSB window $w_{PSB}/h \sim 7.8 \pm 1.2$ GHz is comparable to $\Delta_L/h \sim 5.48$ GHz used for the simulation of the energy diagram in the odd configuration in Fig. 6e. In our system with a magnetic field $B \ll 1$ mT, the effect of the spin-orbit interaction or Zeeman splitting difference between the two QDs can be neglected[10,53]. Consequently, $t_{12}, t_{21} \sim 0$ in the model Hamiltonian, and the resonator electric field can only drive spin-preserving transitions. Furthermore, in the "(1,1)" configuration, the spatial separation of the two holes results in a negligible exchange splitting between the "(1,1)g" and "(1,1)e" states (Fig. 6d), allowing us to set $\Delta_R \sim 0$ in the model Hamiltonian. This leads to the energy diagram depicted in Fig. 6f, which explains the observation of a conventional charge qubit-like spectrum reported in Fig. 6b, corresponding to the charge transition "(2, 0)g"↔"(1, 1)g" with excitation energy $\Delta E_2$. We also demonstrate that the master equation model constructed using our extended effective Hamiltonians closely reproduces the measured spectra of the hybridized multi-level DQD-resonator system (Fig. 6a, b) for both odd and even configurations, as reported in Supplementary Note 10.

As a side note, we observe faint additional features in some of our 2D spectroscopy datasets, such as the ones around 5 GHz in Fig. 6a. These features may be attributed to uncontrolled two-level fluctuators in the tunneling junctions of the SQUID array resonator, which can capacitively couple to the microwave photons (see Supplementary Note 3)[60]. Alternatively, transitions between higher DQD energy levels, which are observable due to a finite thermal population of the excited state, might also explain some of these extra-avoided crossings. However, accurately modeling these transitions would require the

introduction of additional energy states into our model, which is beyond the scope of this work.

## Discussion

In this study, we have demonstrated the potential of a hybrid architecture, combining a superconductor cavity with semiconductor QDs for advancing hole-based quantum information processing in planar germanium. Leveraging a high-impedance Josephson junction-based resonator with tunable frequency, we have demonstrated strong hole charge-photon coupling. This achievement is substantiated by our observation of charge-photon vacuum-Rabi mode splitting and the high cooperativity value ($C \sim 100$) estimated for our hybrid system. Furthermore, the frequency tunability of our resonator has enabled us to successfully resolve SCSs within QDs in planar Ge structures. The distinct spin symmetries of the SCSs lead to significantly reduced decoherence rates of the higher excited levels, a promising development for establishing strong spin-photon coupling. The interaction between QD SCSs and a frequency-tunable resonator provides a very effective avenue for exploring complex many-body electronic states in multi-level QDs. While a detailed measurement of the charge density distribution of the ground state is required to unambiguously prove the Wigner molecularization process[44], the presence of strong Coulomb correlation and QD confinement anisotropy, as suggested by our simulations, make WMs the most plausible model to describe the observed quenching of SCSs in our QDs[38,45–50]. Our findings facilitate coherent photon coupling with spin-charge hybrid qubits, also potentially based on longitudinal interaction through singlet-triplet splitting modulation[61]. In conclusion, we have demonstrated the ability to coherently exchange a photon with holes in planar Ge, marking a critical step toward achieving long-distance spin-spin entanglement. Our work lays the foundation for future research on hole-photon coupling and long-range interactions of hole-based qubits, paving the way for the development of large-scale quantum processors.

## Methods

### Device fabrication

The hybrid triple QD device is fabricated on a Ge/SiGe heterostructure grown by low-energy plasma-enhanced chemical vapor deposition (LEPECVD) using a forward grading technique (see Fig. 1b)[53]. The device fabrication is entirely carried out at the Center of MicroNano Technology (CMi) at EPFL. As a first step, 60-nm Pt markers and ohmic contacts are patterned by E-beam lithography (EBL), evaporation, and lift-off. Immediately before the deposition, a 20-s dip in diluted HF (1%) removes the native oxide in the opened regions to ensure a low-resistive ohmic contact. The 2DHG is self-accumulated in the 16 nm Ge QW. Therefore, a 110 s reactive ion etching (RIE) step etches ≈80–90 nm, leaving a well-defined conductive channel from one ohmic contact to the other. The reacting plasma is based on $SF_6$, $CHF_3$, and $O_2$ and the mask is patterned by EBL. A 15-s dip in buffered HF etches away the native oxide immediately before the gate oxide deposition, a 20 nm atomic layer deposition (ALD) $Al_2O_3$. The deposition temperature is 300 °C. Then, a 15-minute rapid thermal annealing (RTA) in forming gas ($N_2/H_2$ 5%) at 300 °C ensures that the Pt properly diffuses down to the Ge QW. The single-layer gates are patterned in two steps by EBL, evaporation and lift-off. This ensures that the thin 3/22 nm Ti/Pd gates are patched on the etched step by 3/97 nm ones, routed out to the bonding pads. The superconducting part of the device is again patterned in two steps by EBL, evaporation and lift-off. First, the waveguide and the ground plane (120 nm of Al) and, lastly, the SQUID array resonator, following the conventional Dolan-bridge double angle evaporation method for Josephson junctions (JJs). The bottom Al layer is 35-nm thick, whereas the top one is 130 nm. The tunneling oxide barrier is grown by filling the chamber with $O_2$ at a pressure of 2 Torr for 20 min (static oxidation) without breaking the

vacuum. From measurements of the SQUID array resistance at room temperature, we estimate a critical current of about 522 nA per SQUID.

## Fitting procedure for a conventional cavity-dressed charge qubit

The experimental data shown in this work reporting the feedline transmission $S_{21}$ are fitted to a master equation model (see Supplementary Note 4 for the full derivation) and normalized by a background trace to remove the standing wave pattern present in the feedline transmission. The background reference trace is obtained by tuning the resonance frequency of the resonator outside the frequency region of interest by making use of the superconducting coil and recording a high-power trace. The complex transmission of a resonator hanged to a 50 Ω feedline and coupled to a charge qubit reads as:

$$S_{21} = ae^{i\alpha}e^{-2\pi i f_d \tau}\frac{\Delta_r - i(\kappa - |\kappa_{ext}|e^{i\phi})/2 + g_{eff}\chi}{\Delta_r - i\kappa/2 + g_{eff}\chi}, \quad (1)$$

where $\Delta_r = \omega_r - \omega_d$ is the resonator-drive detuning, $\kappa = \kappa_{ext} + \kappa_{int}$ the total resonator linewidth given by both coupling to the waveguide $\kappa_{ext} = |\kappa_{ext}|e^{i\phi}$ and internal losses $\kappa_{int}$, $g_{eff} = g_0\frac{2t_c}{\sqrt{\varepsilon^2 + 4t_c^2}}$ the effective charge-photon coupling strength, $\varepsilon$ the DQD detuning, $t_c$ the inter-dot tunneling coupling, $\chi = \frac{g_{eff}}{-\Delta_q + i\Gamma}$ the DQD susceptibility, $\Delta_q = \omega_q - \omega_d$ the qubit-drive detuning, with the qubit frequency $\omega_q/2\pi = \sqrt{\varepsilon^2 + 4t_c^2}/h$, and $\Gamma$ the charge qubit linewidth. $a$, $\alpha$, $\tau$, and $\phi$ are correction factors that take into account the non-ideal response of the cavity due to the environment. Further information is provided in Supplementary Note 4. The resonator parameters $f_r$, $\kappa$, and $\kappa_{ext}$, as well as the environmental factors, are obtained by separately fitting $S_{21}$ for the bare uncoupled resonator.

The simultaneous fit of the line-cuts reported in Fig. 2d and Supplementary Fig. 4g–i is performed using common fitting parameters for $t_c$ and $\Gamma$, while using separate $g_0^k$, where $k$ = g, h, i for the three different datasets. In order to convert the voltage axis to DQD detuning $\varepsilon = \mu_L - \mu_R = \beta_{pL}V_{pL} - \beta_{pR}V_{pR}$, the differential lever arms $\beta_{pL} = \alpha_{pL}^L - \alpha_{pR}^R = 0.031$ and $\beta_{pR} = \alpha_{pR}^R - \alpha_{pR}^L = 0.016$ are extracted from Coulomb diamond and DQD charge stability diagram measurements. Here, $\mu_L$ ($\mu_R$) is the electrochemical potential of the left (right) QD, $\alpha_{pL}^L$ ($\alpha_{pR}^R$) is the lever arm for the left (right) plunger gate and $\alpha_{pL}^R$ ($\alpha_{pR}^L$) is the cross-lever arm for the left (right) plunger gate.

Figure 3c is obtained from a separate measurement with respect to Fig. 3a, with higher resolution and integration time, and taken at a slightly different flux point ($V_{flux} = 507$ mV). The resonator parameters used for generating Fig. 3d are obtained from fitting the bare resonator as a function of $V_{flux}$, similar to the measurement in Fig. 1g (see Supplementary Fig. 6). The other parameters are obtained from fitting a frequency line-cut of Fig. 3b at $V_{flux} = 504$ mV (see Supplementary Fig. 8b), where the two subsystems are in resonance, to the master equation model described above. Note that here, the lowest frequency of the measurement is 3.8 GHz, limited by the bandwidth of the cryogenic circulators. To help interpret the different extracted $g_0$ for the datasets in Fig. 3a and Supplementary Fig. 9, the resonator differential lever arm is calculated for both cases, following the relation $g_0 = \frac{1}{2}\beta_r V_{0,rms}/\hbar = \frac{1}{2}\beta_r 2\pi f_r\sqrt{\frac{Z_r}{2\hbar}}$ [32]. Using $Z_r = 1.6$ kΩ (1.2 kΩ) and $f_r = 4.149$ GHz (5.432 GHz) for Fig. 3a (Supplementary Fig. 9a), we get $V_{0,rms} = 7.6$ μV (8.7 μV) and $\beta_r = 0.18$ eV/V (0.25 eV/V). For further details about the resonator equivalent lumped impedance and its frequency, see Supplementary Notes 3.

In contrast to Fig. 3, the fits presented in Fig. 4 and Supplementary Fig. 10 are performed simultaneously on all the 2D spectroscopy datasets. To account for the detuning dependence of the qubit dephasing rate due to charge noise, we include a DQD detuning

dependence of the qubit decoherence in the form of $\Gamma = \Gamma_0 + \Gamma_\varepsilon\frac{1}{\hbar}\frac{\partial\omega_q}{\partial\varepsilon} = \Gamma_0 + \Gamma_\varepsilon\frac{\varepsilon}{\hbar\omega_q}$ [23,62,63], where the derivative $\frac{\partial\omega_q}{\partial\varepsilon}$ quantifies the sensitivity of the qubit energy, and hence the scaling of the qubit dephasing rate, with respect to detuning noise induced by charge noise in the environment. For the combined fit, the DQD tunnel coupling $t_c$, the differential lever arm $\beta_{pL}$ of the left plunger gate as well as the constant and detuning-dependent decoherence coefficients $\Gamma_0$ and $\Gamma_\varepsilon$, respectively, are shared among all five datasets, while the charge-photon coupling strength $g_0$, a voltage offset $V_{pL}^0$ that corresponds to $\varepsilon = 0$ and all the resonator parameters are fitted independently for each dataset. $\Gamma_0$ was fixed to 57 MHz, extracted from the fit in Fig. 3c. However, we verified that this does not have any influence on the estimated values for $g_0$. We extract a value of $\Gamma_\varepsilon \sim 164$ MHz from the fit. All resonator parameters (see Eq. (1)), except for the bare resonator frequency $f_r$, are estimated by fitting a single trace of $S_{21}$ taken at large DQD detuning $\varepsilon$. The error bars in Fig. 4e, f correspond to the $2\sigma$ confidence interval estimated by propagating the errors of $\kappa$ and $\kappa_{ext}$ taken from the separate resonator fit. The dashed line in Fig. 4e represents a fit (including the errors of $g_0$) of the extracted $g_0$ values to the relation $g_0 = a\cdot\sqrt{f_r}$. The resulting evolution of $g_0$ is then converted to $g_{eff} = g_0 2t_c/f_r$ and reported in Fig. 4f. The shaded regions in Fig. 4e, f correspond to the $2\sigma$ confidence interval extracted from the last fit above.

## Hamiltonian for SCS simulation

To numerically reproduce the hybridized DQD-resonator spectra obtained from the microwave feedline transmission shown in Figs. 5 and 6, Supplementary Note 8 and Supplementary Note 10, we need, as a first step, to identify the multi-level energy spectra characterizing the DQD in each configuration.

We model the DQD assuming a 4 × 4 toy-model Hamiltonian identical to the spin-charge hybrid qubit defined by three particles in a DQD [52,57], as reported below. The Hamiltonian is written in the position basis $[|L_g\rangle, |L_e\rangle, |R_g\rangle, |R_e\rangle]$, where L (R) denotes the charge state with the excess hole in the left (right) QD. g and e present the ground and excited states of the corresponding charge configuration, respectively. Specifically, in the case of an odd total number of holes in the DQD, L = (2n + 2, 2m + 1) and R = (2n + 1, 2m + 2). Here, we use the notation (p, q) to denote the DQD charge number configuration, with p (q) representing the number of holes in the left (right) QD. Throughout this work, the 2n (2m) core holes in the left (right) QD play no role, reducing the effective DQD charge number to L = "(2, 1)" and R = "(1, 2)", respectively. Similarly, in the even configuration, the relevant charge states effectively become L = "(2, 0)", and R = "(1, 1)". Consequently, the basis in which the Hamiltonian is expressed is ["(2, 1)g", "(2, 1)e", "(1, 2)g", "(1, 2)e"] in the odd configuration and ["(2, 0)g", "(2, 0)e", "(1, 1)g", "(1, 1)e"] in the even one (see Fig. 6c, d for the schematic visualization of these states). In this basis, the DQD 4 × 4 Hamiltonian reads:

$$H = \begin{bmatrix} \varepsilon/2 & 0 & t_{11} & t_{12} \\ 0 & \eta_L\varepsilon/2 + \Delta_L & t_{21} & t_{22} \\ t_{11} & t_{21} & -\varepsilon/2 & 0 \\ t_{12} & t_{22} & 0 & -\eta_R\varepsilon/2 + \Delta_R \end{bmatrix} \quad (2)$$

Here, $\varepsilon$ is the DQD detuning, $\Delta_L$ ($\Delta_R$) is the singlet-triplet splitting $\Delta_{ST}$ when two holes are paired in the left (right) QD, and $t_{ij}$ denotes the tunnel coupling between the $i$th state of left QD and $j$th state of right QD. We also include $\eta_L = 0.92$ ($\eta_R = 0.913$) to account for the different lever arms of the excited states in Fig. 5 [47].

The Hamiltonian eigenvalues are used to reconstruct the energy spectra (eigenenergies vs $\varepsilon/h$) reported in Figs. 5d and 6e, f (top panels), whereas the excitation spectra, i.e., the energy differences between excited states and the ground state, are displayed in the bottom respective panels.

In the odd total hole number configuration, "(2, 1)k"↔"(1, 2)k", with $k$ = g, e (Fig. 5 and Fig. 6a), the ground and the first excited states have the same total spin number $S_{tot}$ = 1/2. For example, "(2, 1)g" and "(2, 1)e" form the doublet spin states with an energy splitting given by the exchange interaction of the paired holes in the left QD[52]. Thereby, a finite tunnel coupling between ground and exchange-split excited states are allowed, e.g., $t_{12}$, $t_{21}$ > 0, by spin-selectrion rules[52]. Both the QDs can have $\Delta_{ST}/h$ ~5 GHz (close to resonator frequency) when an SCS is formed in each QD.

In contrast, in the even configuration, "(2, 0)k"↔"(1, 1)k" (Fig. 6b), the ground ($S_{tot}$ = 0) and first excited ($S_{tot}$ = 1) states do not have the same spin quantum number[58]. For this reason, the terms $t_{12}$, $t_{21}$ ~0, the transition rates for "(2, 0)e"↔"(1, 1)g" (corresponding to "(2, 0)T"↔"(1, 1)S") and "(2, 0)g"↔"(1, 1)e" (corresponding to "(2, 0)S"↔"(1, 1)T") are negligible in our setup with $B \ll 1$ mT (see Fig. 6d)[53,64]. Also, spatial separation of the holes in the "(1, 1)" configuration, results in a negligible exchange splitting between "(1, 1)g" and "(1, 1)e", and we set $\Delta_R$ = 0 in model Hamiltonian for the even case. Because the number of holes in the left QD is the same as in the odd case, we keep the same value of $\Delta_L$ as in the odd case.

The aforementioned Hamiltonian, combined with the generalized input-output theory for a multi-level DQD system interacting with a superconducting resonator (see Supplementary Note 7)[59], reproduces the features observed in the panels of Figs. 5a and 6a, b. The relevant Hamiltonian parameters are shown in Supplementary Notes 8 and 10.

It is worth noting that a similar model can be applied to investigate a resonator coupled to a generic multi-level DQD system, thus offering opportunities to explore valley-orbit states in silicon coupled to superconducting resonators[65].

## Data availability
The datasets generated during the current study are available in Zenodo with the identifier https://doi.org/10.5281/zenodo.13935167.

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

## Acknowledgements

The authors thank Simone Frasca, Vincent Jouanny, Guillaume Beaulieu, Camille Roy, Dominic Dahinden, Davide Lombardo, Daniel Chrastina, and Siddhart Gautam for contributing to some cleanroom fabrication steps, the measurement setup, device simulations, data analysis, and for the useful discussions. P.S. acknowledges support from the Swiss National Science Foundation (SNSF) through the grants Ref. No. 200021 200418 and Ref. No. 206021_205335, and from the Swiss State Secretariat for Education, Research and Innovation (SERI) under contract number 01042765 SEFRI MB22.00081. W.J. acknowledges support from the EPFL QSE Postdoctoral Fellowship Grant. S.B., D.L., and P.S. acknowledge support from the NCCR Spin Qubit in Silicon (NCCR-SPIN) Grant No. 51NF40-180604. M.J., G.K., G.I., and S.C. acknowledge support from the Horizon Europe Project IGNITE ID 101070193. G.K. acknowledges support from the FWF via the P32235 and I05060 projects.

## Author contributions

F.D.P., F.O., W.J., and P.S. conceived the project and wrote the manuscript with inputs from the authors. F.D.P. and F.O. fabricated the device built the experimental setup, and developed the fabrication recipe with inputs from M.J. and G.K. F.D.P., F.O., and M.J. designed the hybrid device. F.D.P., F.O., and W.J. performed the electrical measurements, analyzed the data, and contributed equally to this work. W.J. and S.B. derived the theoretical model and simulated the SCSs. G.I. and S.C. designed the SiGe heterostructure and performed the growth. P.S., D.L., and G.K. initiated the project. P.S. and D.L. supervised the project.

## Competing interests

The authors declare no competing interests.
