## [Transparent Peer Review file · Nature Communications]

Strong hole-photon coupling in planar Ge for probing charge degree and strongly-correlated states

Corresponding Author: Professor Pasquale Scarlino

Version 1:

Reviewer comments:

Reviewer #1

(Remarks to the Author)

The manuscript by De Palma, Oppliger and Jang et al. reports on the first realization of strong dipole coupling of a hole double quantum dot (DQD) in planar germanium and a microwave photon in a superconducting resonator. The hybrid circuit quantum electrodynamics (cQED) device is well within the strong coupling regime with coupling strengths of about 260 MHz and cooperativities reaching a value of 100. In addition, the high impedance tunable resonator is used to perform spectroscopy on the double quantum dot system, revealing standard charge qubit properties for an even charge configuration and low energy excited states for the odd charge configuration. The authors claim that these low-lying excited states originate from Wigner molecular (WM) states, largely favored by anisotropic confinement potentials in the quantum dots formed in planar germanium quantum wells.

The measurements are carried out with care and the analysis of the data is overall sound. Some points need to be clarified on the definition of g_0 and g_{eff} as pointed out below.

Even though the authors convincingly demonstrate that there are low energy excited states present in the DQD system, there is no evidence or proof that these states are due to WM states (even if it might be the case). Indeed, a debate could be opened here: one could argue that any multi-charge quantum dot is experiencing a renormalization of the orbital energy level due to interaction but is this sufficient to speak about WM states? A many hole quantum dot will by default host a much denser energy spectrum than in the single charge regime. For example, the spectrum presented in FigS4 clearly shows quasi-degenerate eigenstates which could explain the spectra measured experimentally.

Some of the features in the datasets presented in Fig. 5a and 6a are reproduced by the toy model invoked to describe WM states. However, there are clearly indications of other states (for example Fig. 6a third panel from the top where there are at least two additional states visible in the lower energy polariton branch), that are not captured with the toy model and which seem to be even at lower energy. How do the authors explain that?

Furthermore, the claim that transitions between some of these states are more coherent due to their spin structure is not supported by the data as the exact spin structure of the states is unknown (all measurements are performed at zero magnetic field).

Fig. 6 b and f show an energy spectrum of the DQD with multiple states for which the authors do not provide convincing experimental data. For example, Δ_R could range from 0 (as assumed in the manuscript) up to Δ_L without changing the measurement. Furthermore, the blue transition cannot be easily probed with the resonator as it does not involve a large charge dipole (" $(2,0)g \leftrightarrow (2,0)e$ "). How do the authors know that this transition and hence the excited states exist in this DQD configuration?

Considering the above arguments, the authors do not present a convincing argument that the low energy states are indeed due to WM states. Therefore, the manuscript reduces on the report of a strong charge-photon coupling with the detection of additional low energy states. The spectroscopy of the QDQ states by varying the resonator frequency is nice, but not novel as it has been already reported in Ref. 30 by parts of the authors. The manuscript, therefore, does not meet the criteria for publication in Nature Communications.

Minor comments:

Confusing notation of g_0 and g_{eff} throughout the paper:

There exist several definitions of g_0 and g_{eff} throughout the paper. The fact that the resonator is frequency tunable adds another degree of complexity. However, the readers would clearly benefit from a concise notation throughout the paper. As mentioned in the methods section "Fitting procedure for a conventional cavity-dressed charge qubit", the charge photon coupling $g_{\text{eff}} = g_0 \cdot 2t / (\sqrt{4t^2 + \epsilon})$, with $g_0 = 0.5 \cdot \beta_r \cdot V_{0,\text{rms}} / \hbar$ is correctly defined for a fixed frequency. In this case, g_0 will naturally scale with the frequency of the resonator. However, the authors use a different convention for g_0 and g_{eff} in section 4 of the paper where they analyze the data of Fig. 4. Could the authors please converge on a consistent notation? In the same analysis, the authors vary the differential lever arm β for the different cases in Fig. 4a. How do they justify that the lever arm changes for the same DQD configuration?

Side note to that: The resonance frequencies quoted in section 7 "Fitting procedure for a conventional cavity-dressed charge qubit" concerning dataset of Fig. 3a and Fig 3d (at the bottom of page 12) do not correspond to the resonance frequencies in the corresponding figures. Could the authors please correct that?

Point-by-point list:

- Could the authors comment on the number of charges in the DQD system as this is highly relevant for their claim on the WM states.
- In the opinion of the referee Fig. 2 e – I do not add any value as the same data is a lot more convincingly shown in Fig. 4. In addition, Fig. 3 does not add much value and could be moved to the supplementary. The presentation of all these datasets renders the manuscript lengthy to read.
- Could the authors comment if the source drain bias applied to measure Fig. 2 a is also applied to measure Fig. 2 b and c?
- The devices used in Ref. 32 are not silicon finFETs as wrongly stated in the introduction but rather silicon nanowire transistors.
- Why do the authors fit the 2D dataset instead of a line cut (e.g. Fig 4b)? Is it justified to assume a constant decoherence rate γ as a function of ϵ ?

Reviewer #2

(Remarks to the Author)

This manuscript reports two advances in planar Ge double-quantum-dot (DQD) heterostructures that have emerged as frontrunners for future hole-charge quantum qubits: (1) experimental realization of a strong coupling with microwave photons in a superconducting resonator, and (2) utilization of the frequency tunability of the resonator to explore in detail the quenched energy splitting associated with strongly-correlated Wigner molecule (WM) states.

Apart of the technical achievements in point (1), the explicit discovery of strong WM physics in Ge DQDs [point (2)], coming close on the heels of similar discoveries in GaAs and Si quantum-qubit devices, provides a glaring example of the synergy between fundamental many-body physics and applied physics; it will motivate further theoretical and experimental studies.

I have only a couple of small revisions to suggest concerning section S4 of the supplemental notes:

- (i) This section provides useful ingredients for getting one's hands on the quenching of the Ge-QD energy spectrum. Fig. S4 (spectrum as a function of anisotropy) is very useful. In this context, the paper P.S. Drouvelis, P. Schmelcher, and F.K. Diakonov, Phys. Rev. B 69, 035333 (2004) is worth citing; it investigated the spectrum of a two-electron elliptic QD as a function of anisotropy using the method of separation of center-of-mass and relative coordinates.
- (ii) The method of separation of center-of-mass and relative coordinates is applicable only to single QDs with harmonic confinements. For a DQD, more elaborate methods are needed, as was done in Ref. [9] of the supplementary notes. This fact should be mentioned explicitly.

Recommendation: With the two small revisions mentioned above, I gladly recommend acceptance.

Reviewer #3

(Remarks to the Author)

The authors demonstrate strong hole-photon coupling with a coupling strength up to 260 MHz. The photons are excited in a quarter-wave high-impedance SQUID array cavity, the resonance of which can be tuned by a local magnetic field coil. Holes are confined in a planar Ge/SiGe double quantum dot. They can form Wigner molecules with excitation energy matching the resonator frequency and spin splitting without external magnetic fields. Latter could solve a conflict of hybrid integration concerning the magnetic field required for qubits on the one hand and SQUID arrays on the other hand.

The authors measure stability diagrams of a hole double quantum dot and extract the energy spectrum of the highly-correlated Wigner molecules with various hole-fillings and detuning using the resonator as a detector.

The authors present measured data of high quality. In general, the manuscript is very clear and well written and I appreciate the many details in the Methods and the Supplementary Sections. Germanium is a promising platform for the realisation of spin qubits. A remaining challenge is the scaling and long-range entanglement of spin (-charge) qubits. The presented strong coupling between a hole charge-qubit and a cavity is therefore an important first step. However, the strong coupling to a spin-qubit and the coherent coupling of two spin qubits are essential for the target of qubit entanglement via a cavity, but not presented here. The strong coupling was already demonstrated in reference [32] for hole spin qubits. Coherent spin-qubit coupling and entanglement was already demonstrated for electron spin qubits [<https://doi.org/10.48550/arXiv.2310.16805>]. The energy spectroscopy of spin-charge states by hybrid cQED has been shown as well. [<https://doi.org/10.1103/PhysRevLett.119.176803>]. Therefore, the significance of the presented measurements should be sharpened and better categorised in the overall context for publication in Nature Communications. Perhaps the manuscript should appear in a more specialised journal.

Here some additional points

1. Please comment on the number of holes occupying the double quantum dot.
2. To clarify the impact, please comment on how the observed Wigner molecules can be used as qubits. Will their performance be compatible with demonstrated hole-spin qubits in Ge?
3. Can the energy spectrum of Wigner molecules be reproduced in many Ge dots for perspective of entanglement?
4. Please elaborate on the significance to use the presented method for “in-depth studies of strongly correlated electronics states in open systems”.

Minor points:

5. What is the magnitude of the magnetic field in the double quantum dot generated by the coil?
6. What is the admixture of light-hole in the characterized Wigner molecules?
7. Please label charge states in the figures (as done in Fig. S5).

Version 2:

Reviewer comments:

Reviewer #1

(Remarks to the Author)

The authors have addressed all points raised by the reviewers and made changes to the manuscript and supplementary material wherever necessary. The manuscript has significantly improved in readability and is now much easier to follow and understand. I especially appreciate the added information on additional states interacting with the resonator that have not been discussed before (e.g. two-level systems in the SQUID-array). Therefore, the manuscript in its current form is sound and worth publishing.

To conclude, the overall message of the manuscript can be summarized as follows:

- 1) first demonstration of strong charge – photon coupling for hole carriers in planar Ge structures
- 2) investigation of low-lying energy states in multi-charge quantum dots by resonant spectroscopy taking advantage of a frequency tunable resonator

I fully acknowledge the achievement of point 1) as an important demonstration for the quantum dot and spin qubit community as planar germanium is a very promising platform for future experiments. The investigation of low-lying energy states by resonant spectroscopy is on its own an interesting technique. However, the presence of such states in multi-charge quantum dots is not at all unexpected (see for example previous reports on spin-charge hybrid qubits). Furthermore, it is obvious as well, that some of these transitions can be more coherent (depending on their spin symmetries). Overall, the information on the low-lying energy states the authors extract from their measurements remain limited. Two-tone spectroscopy and magnetic field dependent measurements would clearly shed more light on the physics involved. Therefore, the novelty of the presented work is limited and a more specialized journal might be more appropriate.

Minor comments:

- The authors implemented a detuning dependent decoherence rate of the charge qubit for the fitting ($\gamma = \gamma_0 + \gamma_{\epsilon} \frac{\epsilon}{\hbar\omega_q}$). They do state the value for γ_0 . However, I was unable to find a value for γ_{ϵ} , which would give the reader an idea on the magnitude of the charge noise in these structures. Could the authors give an order of magnitude for this parameter? Does it contribute significantly to decoherence or not?
- Fig2 (b) misses a label for the y-axis (I guess it should be V_{PR})

Reviewer #2

(Remarks to the Author)

Unfortunately, the revisions made have resulted in a degradation of the theoretical treatment in the manuscript. This is centered around the replacement of the term "Wigner molecule" (which is well established and widely used in the literature for the last 25 years) by the term "Strongly Correlated States (SCSs)" [which is indeterminate, e.g., the Laughlin states in the lowest Landau level (high B) and Kondo-effect states are strongly correlated states].

The following points elaborate further and provide clarifications of the statement above:

1) The manuscript attributes the appearance of the SCSs in the planar Ge DQD (at zero or low magnetic field, B) to larger than unity values of the Wigner ratio λ_W , which expresses the ratio of the Coulomb potential energy over the kinetic energy. Independently of the number of electrons (N), large values of λ_W induce electron localization (Wigner conceived the "Wigner crystal" of electrons under the condition that the potential energy dominates the kinetic energy; see [R1] Phys. Rev. 46, 1002 (1934), <https://journals.aps.org/pr/abstract/10.1103/PhysRev.46.1002>

For a small number $N \leq 10$ of confined 2D or quasi-1D electrons, full-configuration-interaction (FCI) and Quantum Monte-Carlo calculations for large λ_W have confirmed the electron localization and formation of Wigner molecules (WMs, finite-size analogs of the Wigner crystal). Electron localization and WM formation for larger N's (up to 30) have been also found by employing unrestricted Hartree-Fock calculations (see, e.g., [R2] PRL 82, 5325 (1999) <https://journals.aps.org/prl/pdf/10.1103/PhysRevLett.82.5325>, and [R3] arXiv:2406.05886, <https://arxiv.org/abs/2406.05886> (for a case with strong deformation).

2) The fact that heavy holes form WMs for $\lambda_W > 1$ has been recently confirmed experimentally via STM imaging of the 2D charge densities in bilayer TMD moiré superlattices; see [R4] Hongyuan Li et al., arXiv:2312.07607, <https://arxiv.org/abs/2312.07607>

3) The fact that WM formation is accompanied by a strong suppression of energy gaps in the excitation spectra is well documented in the literature. For $N=2-5$ 1D electrons, see the converging (as a function of size) energy spectra in [R5] Phys. Rev. B 70, 035401 (2004), <https://journals.aps.org/prb/abstract/10.1103/PhysRevB.70.035401> For $N=4$ electrons in a 2D double QD, see [R6] Phys. Rev. B 80, 045326 (2009), <https://journals.aps.org/prb/abstract/10.1103/PhysRevB.80.045326> See also [R7] Phys. Rev. B 104, 235302 (2021) <https://journals.aps.org/prb/abstract/10.1103/PhysRevB.104.235302>, and Refs. [38-45] in the revised manuscript.

4) Given the current stage of discussion/debate that arose in the context of the refereeing of this manuscript, as written, the short section 6 in the supplementary notes is inadequate. Namely,

(i) The title is confusing. I do not understand the sentence "Orbital state renormalization due to strong correlation". The word "orbital" usually refers to a single-determinant wave function and to a mean-field approach. However, here an exact 2e wave function is used, which cannot be represented by a single determinant alone, a property shared with the FCI wave functions, and in this context "orbital state" is not appropriate. I suggest that the original title of the first submission be restored.

(ii) The fact that the WM is a well-established concept is not conveyed in this section (and thus in the manuscript). I suggest that the single Ref. [18] in the sentence "... which is often referred to as a Wigner molecule [18]" in this section be expanded to a full list reflecting the earlier abundant WM literature, including Refs. [R1-R7] above, Refs. [38-45] in the manuscript, and the additional references below.

Recommendation: The paper represents a solid advancement to the experimental effort toward the understanding and control of planar Ge-based qubits. As is, however, it is not ready for publication. I suggest that major revisions are needed to convey the proper synergy between the Wigner ratio, QD anisotropy, and formation of Wigner molecules/finite Wigner crystallites.

SOME ADDITIONAL WIGNER-MOLECULE LITERATURE

[R8] Eur. Phys. J. D 16, 373-380 (2001)
<https://epjd.epj.org/articles/epjd/abs/2001/10/iss-89/iss-89.html?mb=0>

[R9] Phys. Rev. B 65, 075309 (2002)
<https://journals.aps.org/prb/abstract/10.1103/PhysRevB.65.075309>

[R10] Phys. Rev. B 65, 115312 (2002)
<https://journals.aps.org/prb/abstract/10.1103/PhysRevB.65.115312>

[R11] Phys. Rev. B 67, 045311 (2003)
<https://journals.aps.org/prb/abstract/10.1103/PhysRevB.67.045311>

[R12] Nature Physics 2, 336–340 (2006)
<https://www.nature.com/articles/nphys293>

[R13] Phys. Rev. Lett. 85, 1726 (2000)
<https://journals.aps.org/prl/abstract/10.1103/PhysRevLett.85.1726>

[R14] Phys. Rev. B 77, 035325 (2008)
<https://journals.aps.org/prb/abstract/10.1103/PhysRevB.77.035325>

[R15] J. Chem. Phys. 124, 124102 (2006)
<https://doi.org/10.1063/1.2179418>

Reviewer #3

(Remarks to the Author)

The authors answered well to my points and changed the manuscript accordingly. The rearrangements of figure panels improved the readability. About significance of the work, the authors argue that "planar Ge is currently at the forefront of spin qubit technology, as evidenced by the substantial volume of recent literature utilizing Ge-based hole QDs." I fully agree that planar Ge is one of the promising qubit platforms. In other qubit platforms such as holes in MOS devices or electrons in planar Si quantum wells, strong coupling and even coherent spin-coupling was demonstrated. Therefore, doubts remain whether the presented progress justifies publication in Nature Communications.

Version 3:

Reviewer comments:

Reviewer #2

(Remarks to the Author)

Taking the concerns in my previous report under consideration, the authors have revised the manuscript sufficiently well. The revised version contains a much improved presentation of the theoretical framework. I recommend acceptance.

Reviewer #4

(Remarks to the Author)

Reviewer #5

(Remarks to the Author)

Response to reviewers

We thank the referees for their comments and suggestions, to which detailed responses follow below. Changes to the manuscript are highlighted in red in this document. Additionally, the major changes on the figures are summarized in the following:

Manuscript

Fig. 2 The old panels b, d, f, g and i are removed from the main text. The old Fig. 2 now appears in the supplementary material as Supplementary Fig. 4.

Fig. 3 The old panels c and e are removed from the main text. The line-cut (old panel e) now appears in the supplementary material as Supplementary Fig. 8.

Fig. 4 Panel b now reports the new simultaneous fit. The old panels c and d have been split into c, d and e, f, respectively. The new panels e and f now show the newly extracted values for g_0 and g_{eff} , respectively, as well as the corresponding trend. Also the corresponding DQD detuning values are added to panel f.

Fig. 5 Some labels and colors are slightly adjusted to improve clarity.

Fig. 6 The graphs in panels c and d are rearranged to be easier to interpret. Now, the cases for $\varepsilon < 0$ and $\varepsilon > 0$ are positioned above respective region in panels e and f. Additionally, some labels and colors are slightly adjusted to improve clarity.

Supplementary Material

Supplementary Fig. 4 Newly added figure - Contains the old version of Fig. 2 in order to report the graphs removed from the manuscript.

Supplementary Fig. 6 Newly added figure - Fitted resonator coupling strengths as a function of the resonator frequency.

Supplementary Fig. 7 Newly added figure - Resonator spectroscopy measurements showing spurious modes coupling to the resonator.

Supplementary Fig. 8 Newly added figure - Contains the line-cut of Fig. 3b, which was removed from the manuscript.

Supplementary Fig. 11 Extending the old Supplementary Fig. 4 by adding the radius-dependence of the orbital excitation energy as well as a version of the old Supplementary Fig. 4 without Coulomb interaction ($\lambda_W = 0$). Additionally, the charge density distribution of the ground and the first excited state is shown for both cases.

Supplementary Fig. 13 Some labels and colors are slightly adjusted to improve clarity.

1 Reviewer #1 (Remarks to the Author)

1.1

The manuscript by De Palma, Oppliger and Jang et al. reports on the first realization of strong dipole coupling of a hole double quantum dot (DQD) in planar germanium and a microwave photon in a superconducting resonator. The hybrid circuit quantum electrodynamics (cQED) device is well within the strong coupling regime with coupling strengths of about 260 MHz and cooperativities reaching a value of 100. In addition, the high impedance tunable resonator is used to perform spectroscopy on the double quantum dot system, revealing standard charge qubit properties for an even charge configuration and low energy excited states for the odd charge configuration. The authors claim that these low-lying excited states originate from Wigner molecular (WM) states, largely favored by anisotropic confinement potentials in the quantum dots formed in planar germanium quantum wells. The measurements are carried out with care and the analysis of the data is overall sound. Some points need to be clarified on the definition of g_0 and g_{eff} as pointed out below.

We reply: We appreciate the referee’s precise summary, and acknowledgement of our effort on this work. Including the clarification on g_0 and g_{eff} definitions, we provide detailed point-by-point responses below.

1.2

Even though the authors convincingly demonstrate that there are low energy excited states present in the DQD system, there is no evidence or proof that these states are due to WM states (even if it might be the case). Indeed, a debate could be opened here : one could argue that any multi-charge quantum dot is experiencing a renormalization of the orbital energy level due to interaction but is this sufficient to speak about WM states? A many hole quantum dot will by default host a much denser energy spectrum than in the single charge regime. For example, the spectrum presented in FigS4 clearly shows quasi-degenerate eigenstates which could explain the spectra measured experimentally.

We reply: We are grateful for the referee’s careful and insightful comments regarding the Wigner molecule (WM) states. We fully appreciate and understand the concerns raised and we would like to make use of this opportunity to further elucidate the nature of the excited states observed in our experiments based on our measurements and model.

Before detailing our reasoning that leads us to identify Wigner molecularization as the most probable mechanism responsible for the observed phenomena, we would like to stress that **the exact nature of the observed low-energy excited states is not of vital importance for the significance of our manuscript. Regardless of whether a WM is explicitly formed, it is clear that Coulomb interaction effects are fundamentally influencing the system’s behavior.**

We believe that a significant **novelty of this work lies in the exploration and observation of exchange-coupled quantum dot (QD) states using a tunable high-impedance resonator strongly coupled to a double quantum dot (DQD).** This observation, which depends on the hole number parity of the QDs, is noteworthy regardless of whether the low-lying states are referred to as WMs. Given that the exchange interaction is a general phenomenon in QDs across all materials, our findings have important implications for quantum information science and condensed matter physics, and will be valuable for studying exchange-coupled QD states in general.

Nevertheless we are happy to take this opportunity to clarify and further elucidate the nature of the low-lying excited states observed in our experiments.

The observed QD energy levels are significantly lower than expected orbital splitting ($70 h\text{-GHz}$), which **strongly suggests that Coulomb interactions are profoundly affecting these states.** These gaps cannot be predicted from the lithographic size of our quantum dot layout alone (Fig. R1a). In spite of the uncertainty in the exact number of holes, the distinct measured energy spectrum variations between multiple even and odd configurations, along with observed Pauli spin blockade phenomena, suggest that our system could be effectively described by neglecting the number of “core” holes that do not take part to the QD physics. This approach captures the essential physics through a model that considers only the effective residual holes.

In this regard, we provide a simplified two-body model to describe the Coulomb correlation-driven low-lying energy states. Specifically, when two charged particles are confined within a single QD, they experience competing forces: the confinement potential $E_{\text{orb}} \propto 1/l_{\text{QD}}^2$, which tends to keep the charges at the center of the QD, and the Coulomb repulsion $E_{\text{ee}} \propto 1/l_{\text{QD}}$, which repels the two particles. Here l_{QD} denotes the radius of the QD. In this settings, the Coulomb correlation effect becomes more pronounced for larger QD, as the Wigner-ratio $\lambda_{\text{w}} = E_{\text{ee}}/E_{\text{orb}} \propto l_{\text{QD}}$.

In our QD, we estimate $\lambda_{\text{w}} \sim 4.46$ according to the QD radius $l_{\text{QD}} \sim 70 \text{ nm}$. As can be inferred from Fig. R1b and c, finite λ_{w} readily quenches the orbital splitting down by an order of magnitude which is still higher than our resonator frequency bandwidth ($4 \sim 8 \text{ GHz}$). Here, QD confinement anisotropy α , which is highly-likely to be present in realistic QD devices, brings the energy further down to observable frequency bandwidth. We further plot the expected charge density for both non-interacting and interacting cases for $\alpha = 0.8$ in Fig. R1d and e respectively, where we find the ground state charge density distribution is significantly altered by Coulomb correlation and α , and resembles that of the excited state.

With this simple model, **we successfully introduce the two main ingredients to describe the low-lying energy states, Coulomb correlation and confinement anisotropy, which we believe is also in-line with the referee’s point.** Extending the discussion in this work, one may expect a multi-nodal ground state charge density in a multi-hole QD due to the renormalization instead of the two-node charge density presented in Fig. R1e.

In summary, we have observed low-lying orbital states in Ge QDs coupled to a tunable superconducting resonator, facilitated by exchange interactions within a double quantum dot (DQD), The debate on **whether these strong Coulomb correlation-driven low-energy states can be classified as Wigner molecules is largely semantic and does not detract from the results, nor the novelty and essence of our work.** Our research achieves detailed energy spectroscopy of a multi-state DQD, where **low-energy orbital states**

arise from either the Wigner molecule process or solely through Coulomb interactions among multiple holes within the QDs. In both cases, Coulomb correlations play an essential role.

The diverse energy spectra measured for different charge configurations, notably in even and odd states, alongside signatures of Pauli spin blockade, provide compelling evidence of strong Coulomb correlations. Simulations further corroborate that even minor anisotropies, common in realistic devices, can significantly influence these correlation effects and potentially facilitate Wigner-like processes. Additional evidence indicates that these effects are more pronounced in larger QDs, as demonstrated in Fig. R1.

All evidence collected and simulations conducted support the likelihood of the Wigner molecularization process, yet the fundamental takeaway remains that the presence of low-energy excited states is ultimately due to strong Coulomb interactions that significantly renormalize the orbital energy spectrum.

We stress that this work paves a novel route toward spin-photon interface in DQDs based on exchange interaction exploiting a tunable superconducting resonator. Specifically, we anticipate our work to facilitate, for example, magnetic-field-free realization of spin-photon interfaces, or longitudinal photon coupling to the orbital states which may be interesting in the aspect of quantum information science and condensed matter physics.

We acknowledge that the most definitive evidence for Wigner molecularization would involve measuring the ground state charge density or directly imaging the ground state charge density. However, such measurement is challenging to implement in gate-defined quantum dot (QD) devices. In fact, also previous claims in literature of WMs in gate-defined QDs are typically grounded on the observation of quenched singlet-triplet (ST) or orbital states, which are then compared to simulations of specific QD configurations with a known number of electrons. Similar to other studies, our conclusions are also based on comparisons with simulations that suggest the WM process is likely to occur in our context.

In consideration of the points discussed above, **we have revised our terminology in the manuscript to refer to these states as “quenched orbital states due to a strong Coulomb correlation” or “strongly correlated states” (SCS) rather than “Wigner Molecule (WM) states”,** and modified the manuscript accordingly. Instead, we now describe the Wigner molecularization process as the most likely explanation for our observations.

To clearly present the two main elements: Coulomb-correlation and confinement anisotropy for orbital energy quenching, and their implication on the charge density of the ground state, we now present Fig. R1 in the new version of Supplementary Note 6. We also modified the manuscript as below:

Page 3: The Wigner ratio $\lambda_w = E_{ee}/E_{orb} \propto l_{QD}$ (see Supplementary Note 6) quantifies the ratio between the Coulomb interaction strength ($E_{ee} \propto 1/l_{QD}$) and the orbital confinement energy ($E_{orb} \propto 1/l_{QD}^2$). Coulomb interactions become increasingly relevant in large QDs, as the ones studied here. In our experiment, we estimate $\lambda_w \sim 4.46$. Coulomb correlation renormalizes the energy of orbital states in QDs, thus quenching the orbital splitting and therefore the singlet-triplet splitting Δ_{ST} [5]. Furthermore, anisotropic QD confinement is expected to enhance the correlation effect and reduce Δ_{ST} even further (see Supplementary Note 6) [2], as illustrated in Fig. 1e. Orbital state renormalization induced by Coulomb correlation and confinement anisotropy is expected to significantly alter also the charge density distribution of the ground state, promoting the formation of Wigner molecular (WM) states (see Supplementary Fig. 11) [2, 7, 3, 16, 20].

Figure R1: **Orbital state renormalization due to Coulomb interaction and confinement anisotropy.** **a**, Orbital energy splitting E_1 between ground and first excited state without Coulomb correlation effect as a function of the hole quantum dot (QD) radius. $l_{\text{QD}} \sim 70$ nm denotes the radius of the QD in this work, with the corresponding $\omega_{\text{orb}}/2\pi \sim 70$ GHz. The red shaded region denotes the frequency bandwidth of our resonator 4 – 8 GHz. An effective hole mass of $m^* \sim 0.057m_e$ is assumed, with m_e being the free electron mass. **b** (**c**), E_{rel} as a function of confinement anisotropy α without (with) finite Coulomb correlation effect. A schematic of the confinement potential is shown in **b**, where l_x (l_y) denotes the characteristic length scale of the confinement along the major (minor) axis with $\alpha = (l_y/l_x)^2 \leq 1$. The Wigner ratio $\lambda_{\text{W}} = E_{\text{ee}}/\hbar\omega_{\text{orb}}$ quantifies the electron-electron interaction energy with respect to confinement energy. In **c**, $\lambda_{\text{W}} = 4.46$ corresponds to the expected Wigner ratio in our QD. E_1 (green dots) represents the minimal orbital splitting of the QD corresponding to $l_x > l_y$. **d** (**e**), Charge density function spanned around the center of a QD for non-interacting (interacting) case with $\alpha \sim 0.8$. Driven by Coulomb correlation, the ground state charge density (left panel in **e**) becomes similar to that of the excited state (right panel in **e**).

1.3

Some of the features in the datasets presented in Fig. 5a and 6a are reproduced by the toy model invoked to describe WM states. However, there are clearly indications of other states (for example Fig. 6a third panel from the top where there are at least two additional states visible in the lower energy polariton branch), that are not captured with the toy model and which seem to be even at lower energy. How do the authors explain that?

We reply: We thank the referee’s careful observation into our datasets. Owing to the uncertainty of exact number of holes in the QDs, it is indeed difficult to capture all the dynamics via the simple toy-model employed here. While more detailed investigation is needed to verify the faint and narrow features pointed out by the referee, please note that **already with a simple 4 x 4 Hamiltonian, which assumes a single excited level in each QD, all the relevant features of the experimental data can be successfully captured.**

For the extra features we provide below **two possible explanations: two-level systems (TLSs) residing in the tunnel junctions of the SQUID array cavity and higher level transitions visible by thermal population of the first excited state.**

Figure R2a presents the resonator response as a function of the magnetic flux with all the gate lines grounded, i.e. no charge qubit is defined. With a careful look at the high-resolution spectrum, many avoided crossings can be noticed, as clearly indicated by the orange arrows in the zoom-in reported in Fig. R2b. Due to the small coupling strength and the very tiny linewidth of these transitions, it is not realistic to ascribe any charge qubit nature. **We attribute them to the coupling of the high-impedance resonator to TLSs in the tunneling junctions** [<https://iopscience.iop.org/article/10.1088/1361-6633/ab3a7e>]. These measurements, however, were acquired in a new cooldown and for this reason some of these avoided crossings appear at different frequencies with respect to the performed experiments. We thus report in Fig. R2c, together with the corresponding zoomed-in curves (blue, purple and orange boxes) in Figs. R2d, e and f, the same flux sweep, but taken in the same cooldown of the experiments reported in our work. Here, a charge qubit is hybridizing with the resonator at ~ 5.2 GHz to result in vacuum-Rabi splitting of the resonator mode near the qubit frequency (red dashed line in Fig. R2c, and e). Also in this case, **many extra avoided crossings appear, both above and below the charge qubit frequency. This strongly suggests that these extra transitions cannot be attributed to extra states in the DQD, but most likely they are due to TLSs in the SQUID array resonator.** In the zoom-in of Fig. R2e, three avoided crossings are clearly visible. The two around 5 GHz are the ones also visible in Fig. 6a (third panel from the top) and the highest frequency one is the one also visible in Fig. 4a (top panel). Apparently, the TLS induced avoided-crossings are more clearly visible when the charge qubit frequency is close to the resonator. While the more clear visibility may be simply due to the broadened linewidth of the resonator, lower resonator photon population stemming from the qubit-resonator hybridization may also pronounce the TLS features. This is because at high input power, the TLSs are saturated and can therefore not couple to the resonator mode. A high-power flux sweep, not reported here, shows the disappearance of the avoided crossings, indicating a saturation of the TLSs and further strengthening our hypothesis. We note that these TLS modes can interact with the qubit mediated by the resonator, which may result in a weak DQD detuning dependence in the spectrum.

Furthermore, **finite temperature leads to non-zero excited state population at thermal equilibrium, which may also reveal extra features corresponding to transitions between the higher levels.** As discussed in the Supplementary Note 7, a temperature of ~ 10 mK, which result in negligible first excited orbital state population $P_{\text{thermal}} \sim 0$, is assumed for the simulations presented in this work. Due to the small excited state population at thermal equilibrium, the spectrum mainly reveals the transition between the excited states and the ground state, and the transition from the excited state to higher excited levels are not visible. In contrast, with an effective temperature of ~ 80 mK, the sharp feature (green arrow in Fig. R3b) similar to the one present in the experiments (Fig. R3a) can be reproduced in the resonator spectrum. This is because **the finite thermal population of the first excited state ($P_{\text{thermal}} \sim 0.09$) can reveal the transition between the first excited state and a higher energy level.** For the simulation in Fig. R3b, the same 4×4 Hamiltonian is utilized without assuming extra energy levels, which exhibits the energy dispersion as a function of the DQD detuning shown in Fig. R3c. Specifically, the extra sharp feature corresponds to the transition denoted by green arrow in Fig. R3c, which is the excitation from the first excited branch (blue curve in Fig. R3c) to the highest level (green curve in Fig. R3c). Figure R3d illustrates the corresponding excitation spectrum. While the simple 4×4 model cannot perfectly reproduce the DQD detuning dependence in the experiment, our simulation with higher temperature still demonstrates that transition between higher states may exhibit similar sharp features. In this settings, we fully agree with the referee's point that a large QD may result in a dense set of excited states, and anticipate that including those extra states may allow more accurate modeling of the experiments.

To summarize, we provide two plausible explanations, 1) TLSs in the tunnel junctions of the SQUID array and 2) Higher-level transition due to non-negligible excited state population, for the possible cause of the extra features. We acknowledge that a more elaborated model including extra energy states may enable more accurate simulation, which we believe is beyond the scope of this work. However, we would like to stress that **already a simple toy model can capture most of the features observed in the experimental data. We would like to convey that this has not been explicitly documented or demonstrated in the precedent works. In this context, we strongly believe our model can function as a powerful and universal tool for investigating arbitrary DQD systems in cQED frameworks.** This may be useful for realizing, for example, photon coupling to spin-charge hybrid qubits in DQD, or longitudinal photon coupling schemes. In the modified manuscript, we summarize what discussed above in Supplementary Note 3, where we also add a reduced version of Fig. R2. Moreover, we add to the main text the below sentences to provide possible explanations for the sharp features:

Page 12: **As a side note, we observe faint additional features in some of our 2D spectroscopy datasets, such as the ones around 5 GHz in Fig.6a. These features may be attributed either to uncontrolled two-level fluctuators in the tunneling junctions of the SQUID array resonator, which can capacitively couple to the microwave**

photons (see Supplementary Note 3) [14]. Alternatively, transitions between higher energy levels, which are observable due to a finite thermal population at the excited state, might also explain some of these extra avoided crossings. However, accurately modeling these transitions would require the introduction of additional energy states into our model, which is beyond the scope of this work.

Figure R2: **Observation of the spurious two-level systems in the SQUID array resonator.** **a**, Normalized microwave feedline transmission $|A/A_0|^2$ as a function of the resonator drive frequency f_d and flux line voltage V_{flux} with a charge qubit far detuned from the resonator. The measurement is taken with a low photon number ($n_{ph} \ll 1$). **b**, Zoom-in of the green-box region in **a**. Orange arrows denote the signatures of two-level systems (TLSs) interacting with the resonator. **c**, $|A/A_0|^2$ as a function of f_d and V_{flux} with a charge qubit in resonance with the resonator at ~ 5.2 GHz (red dashed line). **d**, **e**, and **f**, Zoom-in of the orange, purple, and blue box region in **c**. Orange dashed lines denote the signatures of TLSs, irrelevant to the charge qubit. **c** is taken in the same cooldown together with the datasets presented in the manuscript, while **a - b** are taken in a separate cooldown.

Figure R3: **Higher level transitions revealed by finite temperature effect.** **a**, Normalized microwave feedline transmission $|A/A_0|^2$ spanned by resonator drive frequency f_d and DQD detuning ε , identical to Fig. 6a in the main text. **b**, Numerical simulation of $|A/A_0|^2$ with an effective temperature ~ 80 mK. Green arrow denote the extra feature revealed by the finite thermal population of the excited state. **c**, Eigen energy level diagram calculated from the same 4×4 Hamiltonian used in this work. **d**, Excitation spectrum corresponding to transition denoted by blue, orange, and green arrows in **c**. The excitation spectrum are superposed on **a**.

1.4

Furthermore, the claim that transitions between some of these states are more coherent due to their spin structure is not supported by the data as the exact spin structure of the states is unknown (all measurements are performed at zero magnetic field).

We reply: We fully agree with the referee’s point that magnetospectroscopy measurements could better reveal the spin states in general. At the same time, however, we would like to stress that the spin structure of the system can be readily inferred from the measurements presented in this work.

The reasoning presented in the following is:

1. **Spin physics can also be revealed at zero magnetic field, for instance by exchange interaction.**
2. Although the exact spin structure of the ground state remains undefined, the observed PSB, together with the distinct resonator spectroscopy signatures, represent the **validity of the hole number parity-dependent spin physics (even and odd configurations).**
3. This implies that:
 - (a) in the even case, some transitions are forbidden by the spin selection rule (only spin-preserving transitions are allowed and can therefore couple to the resonator);
 - (b) in the odd case, further transitions are enabled by the exchange interaction and are therefore observed in the resonator spectroscopies;
 - (c) These transitions in question are more coherent due to their reduced charge noise susceptibility and reduced relaxation rate of the involved states.

First of all, as clearly demonstrated in multiple previous works [C. X. Yu et al., Nat. Nanotechnol. 18, 741–746 (2023), D. Jirovec et al., Nat. Mater. 20, 1106–1112 (2021), A. C. Johnson et al., Phys. Rev. B 72, 165308 (2005)], an effective hole numbering, which neglects the number of even core holes in QDs, catches the spin dynamics of the multi particle QDs. **In our work, we validate the effective hole number-dependent spin dynamics in the DQD by studying two adjacent inter-dot transitions both via the resonator and in transport**, as shown respectively in Fig. 6 and in Supplementary Note 9. Also related to the next

point (1.5) raised by the referee, **bias-dependent Pauli spin blockade presented in Supplementary Fig. 14 confirms: a) the even hole number occupancy within the DQD studied in Fig. 6b, and b) the presence of an excited orbital state with a distinct spin quantum number from that of the ground state.** Specifically, at negative detuning, effectively two holes occupy the left QD to result in the finite singlet-triplet splitting, exhibiting PSB. However, at positive detuning, the two holes are separated into each QD, resulting in a small exchange splitting between the ground and the excited states. Observation of the PSB, thereby, confirms distinct spin structures between the ground and the excited state and validates our effective hole charge configuration “(2,0)” - “(1,1)”. This also aligns with the resonator spectroscopy of the inter-dot transition (Fig. 6b) which is presenting **a charge qubit-like spectroscopy, taking into account that only the spin-preserving “(2,0)g” - “(1,1)g” transition ($S_{\text{tot}} = 0$) can be driven by the electric field of the resonator.**

Starting from the effective “(2,0)” - “(1,1)” configuration discussed above, **we keep the number of holes in the left QD constant, and deplete only the right QD by one hole to investigate the adjacent inter-dot transition shown in Fig. 6a (odd case).** Thereby, the charge configuration studied in Fig. 6a reduces to an effective “(2,1)” - “(1,2)”. Here, the lowest two energy levels result in spin doublet states with a total spin quantum number $S_{\text{tot}} = 1/2$ [Phys. Rev. Lett. 108, 140503 (2012)]. Specifically, in “(2,1)” charge configuration for example, the ground [excited] doublet state “(2,1)g” [“(2,1)e”] involves the anti-symmetric singlet [symmetric triplet] spin pairing of the two holes in the left QD. Due to the same S_{tot} for “(2,1)g” and “(2,1)e”, such states can be electrically coupled by the exchange interaction, and the transition between them can be driven by the electric field of the resonator. In our study, **because we do not change the number of holes in the left QD, we expect the same orbital level to be present in the left QD, for both odd and even cases, with a similar energy splitting.** This implies that in the left QD the ground and the first excited state still preserve their distinct spin symmetries. We emphasize that the **orbital splitting of the left QD estimated from the spectroscopy of the odd case (Fig. 6a) $\Delta_L \sim 5.48$ h·GHz is in line with the splitting extracted from the PSB, which again supports our modeling.**

Furthermore, the second excitation spectrum (ΔE_2 , orange line in Fig. 5e), which is related to the transition between the ground doublet and excited doublet states discussed above, exhibits smaller linewidth compared to the charge qubit (blue line) in the same DQD configuration. The smaller linewidth clearly suggests a lower decoherence rate, which we attribute to the distinct spin symmetry related to each state. Noting that the effect of the spin-orbit interaction is negligible due to small magnetic field in this work (below 100 μT), the **resonator can only couple the states with the same spin quantum numbers.** In this regard, the fact that this ΔE_2 transition can be probed via the resonator confirms that the relevant states have the same total spin quantum number. Thereby, please note that the energy splitting of these transitions do not change as a function of applied magnetic field due to the same spin quantum number, implying the global magnetic field sweep suggested by the referee may not have an effect on the observed spectrum.

In conclusion, the observed PSB, together with the distinct spectroscopy signatures that we systematically observe for several adjacent DQD configurations, represent the validity of the hole number parity-dependent spin model. The proposed model is also consistent with the observed reduced linewidth in the odd case. Based on above considerations, we hope to have clarified the referee’s concerns about the different spin structures which can be inferred from our datasets. To clearly convey our points, we modified the manuscript accordingly. Below, we highlight the major changes made in the manuscript:

Page 9 - 10: **As we demonstrate in detail below making use of Fig. 6, such effective particle numbering in the QDs readily captures spin structures that depend on the hole number parity [9, 8, 29]. In this regard, we expect distinct spin symmetries in our “(2, 1)” \leftrightarrow “(1, 2)” configuration related to the ground and excited states respectively [21]. More explicitly, we assume the ground (excited) state to involve anti-symmetric singlet “S” (symmetric triplet “T”) spin pairing in the doubly-occupied QD (Fig. 5c). In this configuration, the two lowest energy levels in “(2, 1)” \leftrightarrow “(1, 2)” form doublet spin states together with the single spin in the other QD [21]. For instance, “(2, 1)g” (“(2, 1)e”) forms a doublet state with spin singlet (triplet) pairing in the left QD. Here, finite exchange interaction can couple the ground and excited doublet states [21, 22], because they have the same total spin quantum number $S_{\text{tot}} = 1/2$, despite the different spin symmetries within the doubly-occupied QD. With this exchange interaction, “(2, 1)g” \leftrightarrow “(2, 1)e” or “(1, 2)g” \leftrightarrow “(1, 2)e” transitions can be revealed by our resonator as presented in Fig. 5a, in agreement with the spin selection rule.**

Page 11-12: **Because the number of holes in the left QD is the same as in the configuration presented in Fig. 6a, we expect a similar Δ_L between the “(2,0)g” and “(2,0)e” states. The extracted width of the PSB**

window $w_{\text{PSB}}/h \sim 7.8 \pm 1.2$ GHz is comparable to $\Delta_L/h \sim 5.48$ GHz used for the simulation of the energy diagram in the odd configuration in Fig. 6e.

Page 12: In our system with magnetic field $B \ll 1$ mT, the effect of the spin-orbit interaction or Zeeman splitting difference between the two QDs can be neglected [8, 4]. Consequently, $t_{12}, t_{21} \sim 0$ and the resonator electric field can only drive spin-preserving transitions.

1.5

Fig. 6 b and f show an energy spectrum of the DQD with multiple states for which the authors do not provide convincing experimental data. For example, Δ_R could range from 0 (as assumed in the manuscript) up to Δ_L without changing the measurement. Furthermore, the blue transition cannot be easily probed with the resonator as it does not involve a large charge dipole (“(2,0)g” \leftrightarrow “(2,0)e”). How do the authors know that this transition and hence the excited states exist in this DQD configuration?

We reply: We thank the referee’s detailed observation and comments into our datasets. As discussed already in the response to the above point (1.4), the number of holes in the left QD is the same for the configurations investigated in Fig. 6a and 6b. **Consequently, the same low-lying orbital state exists in the left QD for both configurations.** Therefore, the orbital splitting of $\Delta_L \sim 5.48$ h·GHz is estimated from Fig. 6a. Moreover, the observation of PSB proves the existence of such a low-energy orbital state and the different spin symmetry of the ground and excited state in the left quantum dot (“(2,0)g” and “(2,0)e”).

We understand the referee’s point that in the used model the magnitude of Δ_R can vary without affecting the spectrum. However, **the PSB measurement confirms the effective “(1,1)” charge configuration for positive DQD detuning.** Therefore, in such a configuration, the energy separation between the “(1,1)g” and the “(1,1)e” states is given by the exchange interaction J and the Zeeman splitting difference between the two QDs ΔE_z . **Since in the performed experiments the magnitude of the magnetic field is negligible ($\Delta E_z \sim 0$), and the two holes are spatially separated ($J \sim 0$), we set the “(1,1)g” - “(1,1)e” energy splitting, corresponding to Δ_R in our model Hamiltonian, to 0 accordingly.**

We acknowledge that this may not have been trivial from the manuscript, and added the below sentences in the manuscript:

Page 11-12: Because the number of holes in the left QD is the same as in the configuration presented in Fig. 6a, we expect a similar Δ_L between the “(2,0)g” and “(2,0)e” states. The extracted width of the PSB window $w_{\text{PSB}}/h \sim 7.8 \pm 1.2$ GHz is comparable to $\Delta_L/h \sim 5.48$ GHz used for the simulation of the energy diagram in the odd configuration in Fig. 6e.

Page 12: In our system with magnetic field $B \ll 1$ mT, the effect of the spin-orbit interaction or Zeeman splitting difference between the two QDs can be neglected [8, 4]. Consequently, $t_{12}, t_{21} \sim 0$ and the resonator electric field can only drive spin-preserving transitions.

1.6

Considering the above arguments, the authors do not present a convincing argument that the low energy states are indeed due to WM states. Therefore, the manuscript reduces on the report of a strong charge-photon coupling with the detection of additional low energy states. The spectroscopy of the QDQ states by varying the resonator frequency is nice, but not novel as it has been already reported in Ref. 30 by parts of the authors. The manuscript, therefore, does not meet the criteria for publication in Nature Communications.

We reply: We appreciate the referee’s insightful comments regarding our manuscript, particularly concerning the attribution of observed low-energy excited states in the DQD system to Wigner molecule (WM) states. We acknowledge the complexity in definitively proving that these states are Wigner molecules due to the multi-charge nature of the quantum dots used and the inherent difficulty in pinpointing the exact number of holes.

Nevertheless, we respectfully disagree with the referee’s conclusion concerning the value of our article. Our research addresses a critical milestone within the quantum dot (QD)- based spin qubit community, a field at the forefront of solid-state quantum computing systems. **The primary focus of our manuscript is the achievement of the strong charge-photon coupling regime for QD-based hole systems defined in**

planar Germanium (Ge) and coupled to superconducting resonators. Currently, a substantial portion of the spin qubits community has transitioned to working with holes, predominantly within planar Ge heterostructures. **Planar Ge has played a transformative role in recent advancements in scalability and control strategies for spin qubits, positioning itself as a pivotal platform for both current and future developments.** State of the art spin qubits quantum processor [arXiv:2402.18382v1 (2024), arXiv:2308.02406v1 (2023), Nat. Mater. (2024). <https://doi.org/10.1038/s41563-024-01857-5>, Nat. Nanotech. 19, 21 (2024), Nat. Commun. 14, 3617 (2023), Nature 591, 580–585 (2021), Nat. Commun. 11, 3478 (2020), Nature 577, 487-491 (2020)], with up to 16 QDs [Nat. Nanotech. 19, 21 (2024)], is implemented with planar Ge. In addition, to the best of our knowledge, our work represents **the first successful direct demonstration of strong charge-photon coupling for holes, irrespective of the hosting semiconductor.** Additionally, it is worth highlighting that so far only two other published works have demonstrated strong coupling with charge degrees of electrons (not holes). **This alone represents groundbreaking advancement that has the potential to significantly influence the QD-computing community.**

In addition, as argued in the first part of our reply, we would like to stress that the observed low-energy states, whether labelled as Wigner molecules or otherwise, highlight the fundamental role of Coulomb interactions in shaping the quantum mechanical properties of confined systems. **We leveraged our achievement to conduct an in-depth resonant spectroscopy investigation of a semiconducting multilevel system using our unique tunable high-impedance resonator, significantly enhancing the originality and importance of our research. This study marks the first exploration of “strongly correlated hole states” for multiple holes confined in QDs, very distinct from the approach presented in the literature [Nat. Commun. 14, 2948 (2023)].** The strong coupling regime we attained enables a more straightforward spectroscopic analysis of these strongly correlated states, eliminating the need for spin-to-charge conversion readout schemes and charge sensors. Our investigation is instrumental in gaining a physical comprehension of these strongly correlated systems, with profound implications for readout, initialization, magnetic-free encoding, and photon coupling of spin qubits. **Consequently, our study holds general relevance for understanding the physics of strongly correlated systems, not limited to Ge-based QDs.**

In revising our manuscript, we have chosen to describe the observed low-energy states as “strongly correlated states” rather than “Wigner molecules”. **In summary, whether or not these states are formally recognized as Wigner molecules, it is clear that their development, driven by strong Coulomb interactions, significantly transforms the energy landscape of the QDs, highlighting the importance of our findings. A further innovation of our study is the methodological advancements it introduces, along with detailed spectroscopy of multi-state QDs. This reveals complex dynamics essential for a deeper understanding of QD systems.** Notably, such a level of QD qubit spectroscopy within a circuit quantum electrodynamics (cQED) architecture, especially for hole QD qubits, has not been previously implemented.

We believe that the revisions made in response to the referee’s comments have strengthened the manuscript and clarified the scientific contributions of our work. We once again thank the referee for their constructive critique, which has undeniably enhanced the quality and accuracy of our study.

1.7

Minor comments:

Confusing notation of g_0 and g_{eff} throughout the paper: There exist several definitions of g_0 and g_{eff} throughout the paper. The fact that the resonator is frequency tunable adds another degree of complexity. However, the readers would clearly benefit from a concise notation throughout the paper. As mentioned in the methods section “Fitting procedure for a conventional cavity-dressed charge qubit”, the charge photon coupling $g_{\text{eff}} = g_0 * 2t/(\sqrt{4t^2 + \epsilon^2})$, with $g_0 = 0.5 * \beta_r * V_{0,\text{rms}}/\hbar$ is correctly defined for a fixed frequency. In this case, g_0 will naturally scale with the frequency of the resonator. However, the authors use a different convention for g_0 and g_{eff} in section 4 of the paper where they analyze the data of Fig.4. Could the authors please converge on a consistent notation? In the same analysis, the authors vary the differential lever arm β for the different cases in Fig. 4a. How do they justify that the lever arm changes for the same DQD configuration?

We reply: We deeply appreciate the referee’s comment about the definition of g_0 and g_{eff} . Indeed, the definitions adopted in our manuscript might be misleading. **We are therefore now converging to the notation defined in the methods section “Fitting procedure for a conventional cavity-dressed charge qubit”, where $g_0 = \frac{1}{2}\beta_r V_{0,\text{rms}}/\hbar = \frac{1}{2}\beta_r 2\pi f_r \sqrt{Z_r/(2\hbar)}$ and $g_{\text{eff}} = g_0 2t_c/(\sqrt{\epsilon^2 + 4t_c^2})$.** As the referee pointed out, the frequency tunability of the resonator adds more complexity, including also the frequency dependence of the res-

onator equivalent lumped impedance $Z_r = 1/(2\pi f_r C_r)$, where C_r is the resonator lumped equivalent capacitance, assumed constant over the frequency range. Inserting the frequency dependence of Z_r into the aforementioned definition of g_0 , we get a $\propto \sqrt{f_r}$ increase of g_0 with respect to the resonator frequency. As a consequence, the reduction of the effective coupling strength g_{eff} with the dipole moment $2t_c/(\sqrt{\varepsilon^2 + 4t_c^2}) = 2t_c/(hf_r) \propto 1/f_r$ when the qubit and the resonator are in resonance, is partially damped by the $g_0 \propto \sqrt{f_r}$ increase, leading to a final $\propto 1/\sqrt{f_r}$ dependence of g_{eff} . According to these considerations, we removed the confusing notation of g_0 and g_{eff} from section 4 and we converged to the one reported in the methods section. For what concerns the lever arm β_{pL} changes for the same DQD configuration at different resonator frequencies, we would like to point out that the variations are minimal and come from the fact that each 2D dataset was fitted independently. However, we now solve this issue by fitting all the five 2D datasets of Supplementary Fig. 10 with a common fit, where also β_{pL} is a parameter shared among the five fits (please see the same methods section "Fitting procedure for a conventional cavity-dressed charge qubit"). We also modify the main text as follows:

Page 8: From these spectra, we extract the charge-photon coupling strengths g_0 , and present them as a function of f_r in Fig. 4e. We also estimate the effective charge-photon coupling strengths $g_{\text{eff}} = g_0 2t_c / \sqrt{\varepsilon^2 + 4t_c^2} = g_0 2t_c / hf_r$, when the two systems are in resonance, and report them as a function of both ε and f_r in Fig. 4f. Since $g_0 = \frac{1}{2}\beta_r V_{0,\text{rms}}/\hbar = \frac{1}{2}\beta_r 2\pi f_r \sqrt{Z_r/(2\hbar)}$, where the lumped equivalent resonator impedance can be written in terms of f_r as $Z_r = 1/(2\pi f_r C_r)$, we obtain a frequency dependence of $g_0 \propto \sqrt{f_r}$ and hence $g_{\text{eff}} \propto 1/\sqrt{f_r}$ (assuming constant C_r and at resonance). The dashed line in Fig. 4e represents a fit of the extracted g_0 to the expected frequency dependence. Using this fit, we estimate the evolution of g_{eff} as a function of f_r and illustrate it as a dashed line in Fig. 4f. Notably, the evolution of g_0 does not closely follow the expected trend as a function of f_r . This discrepancy can be attributed to a nonuniform and frequency dependent voltage profile of the resonator mode, potentially due to magnetic flux inhomogeneity along the SQUID array. Alternatively, simultaneous hybridization of the DQD with higher order resonator modes (see Fig. 1g), which approach the qubit frequency in the studied flux range, may influence the coupling to the fundamental mode. Further investigation is required in order to better understand the evolution of g_0 .

Side note to that: The resonance frequencies quoted in section 7 "Fitting procedure for a conventional cavity-dressed charge qubit" concerning dataset of Fig. 3a and Fig 3d (at the bottom of page 12) do not correspond to the resonance frequencies in the corresponding figures. Could the authors please correct that?

We reply: We truly appreciate the referee's careful observation. We now corrected the error.

1.8 Point-by-point list

1.8.1

Could the authors comment on the number of charges in the DQD system as this is highly relevant for their claim on the WM states.

We reply: We thank the referee for the detailed comment on the number of charges. Due to the relatively large size of our dots, they cannot be depleted to single hole regime within a practical gate-voltage range. As a consequence, getting a precise particle number is not straightforward. In this regard, we first present an upper bound of the number based on the saturation carrier density measured when the channel is fully conductive, $p_0 = 7.54 \cdot 10^{11} \text{ cm}^{-2}$. Based on the QD anisotropy ~ 0.8 (Fig. R1c) and QD radius $\sim 70 \text{ nm}$ (extracted from Coulomb diamonds measurements, see Supplementary Note 6), we obtain a QD area $\sim 12000 \text{ nm}^2$, which result in a maximum charge number of ~ 90 . We note, however, that the percolation density of the heterostructure may provide more precise estimate of the charge number within a QD as the QDs are formed when the channel is not fully conductive. While the exact percolation density of our material is not measured, we roughly estimate the percolation density to be $\sim 0.1p_0 - 0.3p_0$ from the previous literature [Mater. Quantum. Technol. 1 011002 (2020), Adv. Funct. Mater. 2019, 29, 1807613 (2019), Phys. Rev. B 79, 235307 (2009), Appl. Phys. Lett. 110, 123505 (2017), Phys. Rev. B 92, 035304 (2015)], which results in a more realistic estimate of the charge number $\sim 9 - 27$.

In conclusion, we would like to stress again the fact that, **even if in this work we are far from the last hole regime, the system can be effectively described with an effective holes number, as also remarked by the observed PSB.** Please notice that also in the work [<https://doi.org/10.1038/s41563-021-01022-2>], on a similar heterostructure, a singlet-triplet qubit has been successfully implemented despite one of the two dots was still far from the last hole regime. This also points toward the direction of a system that can be efficiently described with an effective hole number.

We thank again the referee for the important observation and we thus summarize what discussed above in

Supplementary Note 2.

1.8.2

In the opinion of the referee Fig. 2 e – I do not add any value as the same data is a lot more convincingly shown in Fig. 4. In addition, Fig. 3 does not add much value and could be moved to the supplementary. The presentation of all these datasets renders the manuscript lengthy to read.

We reply: We thank the referee’s detailed suggestions. We now moved Fig. 2d, 2f, 2g, and 2i to Supplementary Note 2, and only display the former 2e, and 2h which is the resonant case. Also, we acknowledge that the previous version of Fig. 3 and the corresponding description of the figure may not have been adding much value into the paper. Thereby, we removed the previous Fig. 3c and Fig. 3e from Fig. 3, and modified the manuscript as below for clarity.

Page 5-6: Thereby, we fix the detuning at $\varepsilon = 0$ and vary the external magnetic flux to fine-tune the resonator frequency f_r into resonance with the qubit frequency $f_q = 2t_c/h$. In Fig. 3b, we report $|A/A_0|^2$ as a function of f_d and flux bias voltage V_{flux} , where the charge-photon hybridization at ~ 4 GHz results in a clear vacuum-Rabi mode splitting. By fitting a line-cut of Fig. 3b taken at $V_{\text{flux}} = 504$ mV (reported in Supplementary Fig. 8b), we extract the parameters $(t_c/h, g_0/2\pi, \Gamma/2\pi) = (2018, 154, 80)$ MHz. These parameters are utilized to numerically reconstruct $|A/A_0|^2$ (Fig. 3d). To better evaluate the cooperativity of our system, in Fig. 3c we report a high-quality vacuum-Rabi mode splitting measured, with increased averaging, as a function of f_d in the same DQD configuration ($\varepsilon = 0$), but at $V_{\text{flux}} = 507$ mV to compensate for a slight drift in qubit frequency.

1.8.3

Could the authors comment if the source drain bias applied to measure Fig. 2 a is also applied to measure Fig. 2 b and c?

We reply: Yes, source drain bias are the same for Figs. 2a, b and c because they are measured simultaneously by recording the dc current and the resonator response at the same time. Please note this has already been stated in the manuscript as reported here below for clarity.

Page 4: Fig. 2a shows a region of the DQD stability diagram spanned by V_{pR} and V_{pL} , measured by probing the dc current through the DQD [26]. To characterize the charge-photon coupling, we simultaneously monitor the feedline transmission at the frequency $f_d = f_r \sim 5$ GHz (see Figs. 2b, c).

1.8.4

The devices used in Ref. 32 are not silicon finFETs as wrongly stated in the introduction but rather silicon nanowire transistors.

We reply: We thank the referee’s detailed look on our paper. We now change the terminology to silicon nanowire transistors as below.

Page 2: While several previous experiments have successfully demonstrated strong coupling for electrons hosted in Si [12, 17, 13], GaAs [19, 23], InAs nanowires [25], and for holes in silicon nanowire transistors [29], the strong coupling of holes in planar Ge has remained elusive [10, 27, 11].

1.8.5

Why do the authors fit the 2D dataset instead of a line cut (e.g. Fig 4b)? Is it justified to assume a constant decoherence rate Γ as a function of ε ?

We reply: We thank the referee’s careful observation on our analysis. The main reason why we fit the full 2D map is to accurately extract the detuning lever arm β_{pL} of the plunger gate directly from these spectroscopy measurements. In fact, we extracted β_{pL} for a nearby DQD configuration still measurable in transport, which may not be the same in the studied configuration. Thereby we believe a full 2D fit may provide more accurate modeling of the experimental data.

Regarding the constant Γ used in our fit, we would like to stress that our main goal here is to extract the coupling strength g_0 . In this respect, we notice that the extracted g_0 is not affected by considering or not a detuning dependence of Γ . Nevertheless, we fully agree with the referee’s point that considering the dependence

of Γ on DQD detuning ε may provide more accurate model for the fitting, and we now present new 2D fits with ε dependent Γ in Fig. 4 in the modified manuscript. We implement a DQD detuning dependence for the qubit decoherence of the form $\Gamma = \Gamma_0 + \Gamma_\varepsilon \frac{1}{\hbar} \frac{\partial \omega_q}{\partial \varepsilon} = \Gamma_0 + \Gamma_\varepsilon \frac{\varepsilon}{\hbar \omega_q}$ similar to [Phys. Rev. Lett. 122, 206802 (2019), npj Quantum Inf 3, 1–4 (2017), Rev. Mod. Phys. 86, 361–418 (2014)] and numerically fitted the 2D datasets in Fig. 4. Here, it is assumed that the charge qubit linewidth is dominated by dephasing due to DQD detuning noise.

To sum up, we now present a 2D fit of all 5 datasets, using shared parameters for t_c , β_{pL} , Γ_0 and Γ_ε which also addresses the previous comment of the referee on the same β_{pL} for different sets. We note that the new model faithfully reproduce the experiments, including smaller visibility of the resonator modes for larger ε in Fig. 4. We have added the below sentences to describe the new fitting method including ε dependence of Γ :

Page 15: In contrast to Fig. 3, the fits presented in Fig. 4 and Supplementary Fig. 10 are performed simultaneously on all the 2D spectroscopy datasets. To account for the detuning dependence of the qubit dephasing rate due to charge noise, we include a DQD detuning dependence of the qubit decoherence in the form of $\Gamma = \Gamma_0 + \Gamma_\varepsilon \frac{1}{\hbar} \frac{\partial \omega_q}{\partial \varepsilon} = \Gamma_0 + \Gamma_\varepsilon \frac{\varepsilon}{\hbar \omega_q}$ [18, 15, 24], where the derivative $\frac{\partial \omega_q}{\partial \varepsilon}$ quantifies the sensitivity of the qubit energy, and hence the scaling of the qubit dephasing rate, with respect to detuning noise induced by charge noise in the environment. For the combined fit, the DQD tunnel coupling t_c , the differential lever arm β_{pL} of the left plunger gate as well as the constant and detuning-dependent decoherence coefficients Γ_0 and Γ_ε , respectively, are shared among all the five datasets, while the charge-photon coupling strength g_0 , a voltage offset V_{pL}^0 that corresponds to $\varepsilon = 0$ and all the resonator parameters are fitted independently for each dataset. Γ_0 was fixed to 57 MHz, extracted from the fit in Fig. 3c. However, we verified that this does not have any influence on the estimated values for g_0 . All resonator parameters (see Eq. (1)), except for the bare resonator frequency f_r , are estimated by fitting a single trace of S_{21} taken at large DQD detuning ε . The error bars in Figs. 4e and f correspond to the 2σ confidence interval estimated by propagating the errors of κ and κ_{ext} taken from the separate resonator fit. The dashed line in Fig. 4e represents a fit (including the errors of g_0) of the extracted g_0 values to the relation $g_0 = a \cdot \sqrt{f_r}$. The resulting evolution of g_0 is then converted to $g_{\text{eff}} = g_0 2t_c / f_r$ and reported in Fig. 4f. The shaded regions in Figs. 4e and f correspond to the 2σ confidence interval extracted from the last fit above.

2 Reviewer #2 (Remarks to the Author)

2.1

This manuscript reports two advances in planar Ge double-quantum-dot (DQD) heterostructures that have emerged as frontrunners for future hole-charge quantum qubits: (1) experimental realization of a strong coupling with microwave photons in a superconducting resonator, and (2) utilization of the frequency tunability of the resonator to explore in detail the quenched energy splitting associated with strongly-correlated Wigner molecule (WM) states.

2.2

Apart of the technical achievements in point (1), the explicit discovery of strong WM physics in Ge DQDs [point (2)], coming close on the heels of similar discoveries in GaAs and Si quantum-qubit devices, provides a glaring example of the synergy between fundamental many-body physics and applied physics; it will motivate further theoretical and experimental studies.

We reply: We truly thank the referee’s precise summary, and acknowledgement of the novelty of our work.

2.3

I have only a couple of small revisions to suggest concerning section S4 of the supplemental notes:

2.3.1

This section provides useful ingredients for getting one’s hands on the quenching of the Ge-QD energy spectrum. Fig. S4 (spectrum as a function of anisotropy) is very useful. In this context, the paper P.S. Drouvelis, P. Schmelcher, and F.K. Diakonov, Phys. Rev. B 69, 035333 (2004) is worth citing; it investigated the spectrum of a two-electron elliptic QD as a function of anisotropy using the method of separation of center-of-mass and relative coordinates.

We reply: We appreciate the referee’s suggestion. We now cite [P.S. Drouvelis, P. Schmelcher, and F.K. Diakonov, Phys. Rev. B 69, 035333 (2004)] as recommended.

2.3.2

The method of separation of center-of-mass and relative coordinates is applicable only to single QDs with harmonic confinements. For a DQD, more elaborate methods are needed, as was done in Ref. [9] of the supplementary notes. This fact should be mentioned explicitly.

We reply: We thank the referee’s detailed suggestion, we now mention the two-dimensional two-center-oscillator (TCO) method in the Supplementary Note 6.

Supplementary Note Page 12: **We note that while the method of COM and relative coordinates well describes the dynamics of a single QD, elaborated models such as two-center-oscillator (TCO) may provide more precise description of the dynamics in a DQD.**

2.4 Recommendation

With the two small revisions mentioned above, I gladly recommend acceptance.

We reply: We truly appreciate the referee’s recommendation for acceptance.

3 Reviewer #3 (Remarks to the Author)

3.1

The authors demonstrate strong hole-photon coupling with a coupling strength up to 260 MHz. The photons are excited in a quarter-wave high-impedance SQUID array cavity, the resonance of which can be tuned by a local magnetic field coil. Holes are confined in a planar Ge/SiGe double quantum dot. They can form Wigner molecules with excitation energy matching the resonator frequency and spin splitting without external magnetic fields. Latter could solve a conflict of hybrid integration concerning the magnetic field required for qubits on the one hand and SQUID arrays on the other hand.

The authors measure stability diagrams of a hole double quantum dot and extract the energy spectrum of the highly-correlated Wigner molecules with various hole-fillings and detuning using the resonator as a detector.

We reply: We thank the referee’s precise summary of our work.

3.2

The authors present measured data of high quality. In general, the manuscript is very clear and well written and I appreciate the many details in the Methods and the Supplementary Sections. Germanium is a promising platform for the realisation of spin qubits. A remaining challenge is the scaling and long-range entanglement of spin (-charge) qubits. The presented strong coupling between a hole charge-qubit and a cavity is therefore an important first step. However, the strong coupling to a spin-qubit and the coherent coupling of two spin qubits are essential for the target of qubit entanglement via a cavity, but not presented here. The strong coupling was already demonstrated in reference [32] for hole spin qubits. Coherent spin-qubit coupling and entanglement was already demonstrated for electron spin qubits [<https://doi.org/10.48550/arXiv.2310.16805>]. The energy spectroscopy of spin-charge states by hybrid cQED has been shown as well. [<https://doi.org/10.1103/PhysRevLett.119.176803>]. Therefore, the significance of the presented measurements should be sharpened and better categorised in the overall context for publication in Nature Communications. Perhaps the manuscript should appear in a more specialised journal.

We reply: We appreciate the referee’s positive remarks on the high quality of our measurements and the detailed presentation in our manuscript. However, as we argue below, we believe our work introduces substantial novelty in several aspects, which may be of broad interest to the physical community.

Indeed, as the referee noted, significant research in the context of hybrid cavity-QDs has been conducted with electron spins in silicon, which we have duly recognized and cited in our manuscript. **However, while electron spins in silicon exhibit relatively long coherence times, they are subject to inherent limitations. These include challenges associated with the valley degree of freedom and the high effective mass of electrons, which complicate the fabrication and control of quantum dots in silicon.** Moreover, the inherently weak spin-orbit interaction in silicon means that electrons cannot be

directly coupled to voltage fluctuations in superconducting resonators. Instead, it necessitates the engineering of magnetic field gradients—an artificial spin-orbit coupling—in the double quantum dot region. This requirement further complicates the realization, control, and reproducibility of such devices.

In addition, the referee accurately observed—as we acknowledge in our manuscript—that Ref. [Nat. Nanotech. 18, 741 (2023)] demonstrated strong coupling between a hole spin qubit in Si and a resonator. However, it’s crucial to emphasize that the semiconductor QD device used in that study features a unique and specialized design: it is a silicon-nanowire-MOS DQD structure confining holes. This specific device geometry enables a significantly large lever-arm interaction with the resonator gate, which greatly enhances the strong coupling to microwave photons. While this development is indeed pioneering and presents significant advancements, it is important to recognize some key limitations of these devices. Specifically, they exhibit limited interdot t_c tunability and, notably, no two-qubit gate has been demonstrated yet. Additionally, these nanowire transistors are exclusively produced by Leti in Grenoble and are predominantly utilized by a limited number of research groups, making them less accessible to the wider scientific community. Therefore, this type of device is not the most widely adopted approach within the global spin qubit community. Instead, the broader research focus is increasingly on confining holes in germanium (Ge), particularly in planar Ge.

Planar Ge is currently at the forefront of spin qubit technology, as evidenced by the substantial volume of recent literature utilizing Ge-based hole QDs. This shift underscores the broader adoption and relevance of germanium in advancing the state of the art in spin qubit research, currently representing the material where the new state of the art of spin qubits have been achieved. Planar Ge has played a transformative role in recent advancements in scalability and control strategies for spin qubits, positioning itself as a pivotal platform for both current and future developments. State of the art spin qubits quantum processor [arXiv:2402.18382v1 (2024), arXiv:2308.02406v1 (2023), Nat. Mater. (2024). <https://doi.org/10.1038/s41563-024-01857-5>, Nat. Nanotech. 19, 21 (2024), Nat. Commun. 14, 3617 (2023), Nature 591, 580–585 (2021), Nat. Commun. 11, 3478 (2020), Nature 577, 487-491 (2020)], with up to 16 QDs spin qubits [Nat. Nanotech. 19, 21 (2024)], is implemented in planar Ge.

Given the significant relevance of the planar Ge platform for the spin qubit community, **our manuscript primarily focuses on achieving the strong coupling regime with superconducting resonators for quantum dot-based hole systems derived from planar Ge heterostructures.** It is also noteworthy that multiple research groups have endeavored to develop high-quality resonators on such heterostructures, which has proven to be quite challenging for reasons not yet fully understood. **In this context, our work constitutes a crucial initial step, demonstrating the compatibility of superconducting resonators with these heterostructures and their strong coupling to a double quantum dot charge degree of freedom.**

Additionally, it is worth highlighting that so far only two other published works have demonstrated strong coupling with charge degrees of electrons (not holes). **To the best of our knowledge, our work represents the first direct demonstration of strong coupling with the charge degree of holes, making our work unique in its achievements. This alone represents groundbreaking advancement that has the potential to significantly influence the QD-computing community.** In comparison to Ref. [Nat. Nanotech. 18, 741 (2023)], it is important to note that in that study, the charge-photon coupling strength was only inferred indirectly through the dispersive shift of the resonator. That approach did not allow for a spectroscopic analysis of the Rabi mode splitting (for charge) due to the limited tunability of the DQD tunneling rates in their system.

Furthermore, the significance of our work not only lies in the demonstration of the strong hole charge-photon coupling, but also in the fact that we reveal and study in details strongly-correlated electronic states within a QD-resonator hybrid cQED framework for the first time. We leveraged our achievement to conduct an in-depth resonant spectroscopy investigation of a semiconducting multilevel system using our unique tunable high-impedance resonator, significantly enhancing the originality and importance of our research. This study marks the first exploration of “strongly correlated hole states” for multiple holes confined in QDs, very distinct from the approach presented in the literature [Nat. Commun. 14, 2948 (2023)]. The strong coupling regime we attained enables a more straightforward spectroscopic analysis of these strongly correlated states, eliminating the need for spin-to-charge conversion readout schemes and charge sensors. Our investigation is instrumental in gaining a physical comprehension of these strongly correlated systems, with profound implications for readout, initialization, magnetic-free encoding, and photon coupling of spin qubits. Consequently, our study holds general relevance for understanding the physics of strongly correlated systems, not limited to Ge-based QDs.

Even though the Hamiltonian which describes the valley states in QDs [Phys. Rev. Lett. 119, 176803 (2017), PRX Quantum 2, 020309 (2021)] may be similar to the one used in our work as the referee points out, it should be noted that the physics explored in our work is fundamentally different from the valley physics in Si QDs. **Specifically, our study probes quenched QD orbital states mediated by exchange interaction (not by valley-orbit coupling) using a tunable superconducting resonator. This is especially important because the exchange interaction in QDs is a general phenomenon observable across all semiconducting materials, suggesting that the observations and analyses from our work could**

be applied more broadly to general QD systems. Therefore, our findings could pave the way for exploring exchange-coupled spin-1/2 systems with microwave resonators, which is of considerable interest for quantum information science and condensed matter physics. Additionally, it is important to highlight that, unlike in the referenced study, we have exploited the tunability of our resonator to explore higher orbital states without needing to elevate the system temperature. This capability underscores the utility of our tunable resonator for investigating quantum states in QDs, which is beneficial not only for quantum information processing but also for exploring complex energy structures within QDs.

Based on these points, we hope to have conveyed that our work introduces strong novelties that could significantly influence the field, extending beyond QD-based quantum information processing to address general condensed matter physics challenges. As recent related work [<https://doi.org/10.48550/arXiv.2310.16805>] appeared during the preparation of our manuscript, we have now explicitly cited this in our revised manuscript along with [<https://doi.org/10.1103/PhysRevLett.119.176803>].

We have also added the below sentences to highlight the significance of our work:

Page 2: **The emergence of SCSs is a general phenomenon, which can take place in QDs defined in any semiconductor platform [16, 6, 3]. Such SCSs have profound implications for quantum information processing, offering an encoding for spin-charge hybrid qubits based on exchange interaction [28, 6].**

3.3 Here some additional points

3.3.1

Please comment on the number of holes occupying the double quantum dot.

We reply: We thank the referee for the detailed comment on the number of charges. Due to the relatively large size of our dots, they cannot be depleted to single hole regime within a practical gate-voltage range. As a consequence, getting a precise particle number is not straightforward. In this regard, we first present an upper bound of the number based on the saturation carrier density measured when the channel is fully conductive, $p_0 = 7.54 \cdot 10^{11} \text{ cm}^{-2}$. Based on the QD anisotropy ~ 0.8 (Fig. R1c) and QD radius $\sim 70 \text{ nm}$ (extracted from Coulomb diamonds measurements, see Supplementary Note 6), we obtain a QD area $\sim 12000 \text{ nm}^2$, which result in a maximum charge number of ~ 90 . We note, however, that the percolation density of the heterostructure may provide more precise estimate of the charge number within a QD as the QDs are formed when the channel is not fully conductive. While the exact percolation density of our material is not measured, we roughly estimate the percolation density to be $\sim 0.1p_0 - 0.3p_0$ from the previous literature [Mater. Quantum. Technol. 1 011002 (2020), Adv. Funct. Mater. 2019, 29, 1807613 (2019), Phys. Rev. B 79, 235307 –(2009), Appl. Phys. Lett. 110, 123505 (2017), Phys. Rev. B 92, 035304 (2015)], **which results in a more realistic estimate of the charge number $\sim 9 - 27$.**

In conclusion, we would like to stress again the fact that, even if in this work we are far from the last hole regime, **the system can be effectively described with an effective holes number, as also remarked by the observed PSB.** Please notice that also in the work [<https://doi.org/10.1038/s41563-021-01022-2>], on a similar heterostructure, a singlet-triplet qubit has been successfully implemented despite one of the two dots was still far from the last hole regime. This also points toward the direction of a system that can be efficiently described with an effective hole number.

We thank again the referee for the important observation and we thus summarize what discussed above in Supplementary Note 2.

3.3.2

To clarify the impact, please comment on how the observed Wigner molecules can be used as qubits. Will their performance be compatible with demonstrated hole-spin qubits in Ge?

We reply: We truly appreciate the referee's suggestion to clarify the impact of our manuscript. **Wigner molecule (WM) states can allow, for example, coherent spin-charge hybrid qubit operations [Nano Lett. 21, 12, 4999 (2021), Phys. Rev. Lett. 108, 140503 (2012)].** Driven by the strong correlation effect, the singlet-triplet splitting can be quenched from $\sim 100 \text{ h}\cdot\text{GHz}$ down to $< 10 \text{ h}\cdot\text{GHz}$ which is well within the typical bandwidth of an experimental setup. **Importantly, the transition between the low-lying states can be driven electrically without requiring a magnetic field aided by the exchange interaction.** While this may increase the susceptibility to charge noise in the system, increasing the decoherence rate compared to typical spin qubits, **magnetic-field-free operation capability is especially useful for interfacing with superconducting circuits.** Moreover, the tunability of the singlet-triplet splitting is more pronounced when the electronic correlation is significant, allowing a highly tunable qubit encoding in a double QD which can electrically couple to a superconducting resonator. This also infers the **possible application in**

longitudinal hole-photon coupling scheme which may be further interesting in the perspective of cavity quantum electrodynamics with quantum dots. To stress the utility of the WM states as a qubit in more detail, we add the below sentence to the manuscript:

Page 2: Such SCSs have profound implications for quantum information processing, offering an encoding for spin-charge hybrid qubits based on exchange interaction [28, 6].

Page 10: The presented Hamiltonian also models a spin-charge hybrid qubit, which can be encoded in the SCSs exhibiting a lower decoherence rate in comparison to a bare charge qubit. Such hybrid qubit also allows all-electrical control of the spin states based on exchange interaction [21, 6].

Page 12: Our findings facilitate coherent photon coupling with spin-charge hybrid qubits, also potentially based on longitudinal interaction through singlet-triplet splitting modulation [1].

3.3.3

Can the energy spectrum of Wigner molecules be reproduced in many Ge dots for perspective of entanglement?

We reply: As reported both in theory [Phys. Rev. B 104, 195305 (2021)] and experiment [Nat. Commun. 14, 2948 (2023)], anisotropic and weak QD confinement can promote Wigner molecularization due to non-negligible electron-electron correlation within the QD. Thereby we expect multiple WMs can be reproduced in a well-designed QD arrays, which can be utilized for encoding and entangling multiple spin-charge hybrid qubits.

3.3.4

Please elaborate on the significance to use the presented method for “in-depth studies of strongly correlated electronics states in open systems”.

We reply: We thank the referee’s detailed comment to strengthen the significance of our work. By the sentence pointed out by the referee, we were hoping to convey that our work facilitates the study of strongly correlated electronic states coupled to microwave photonic systems. However, we fully agree with the referee’s point that we need to be more precise with our claim, and we now tone-down on this argument to provide more concrete example. We have modified the manuscript accordingly as below:

Page 12: Our findings facilitate coherent photon coupling with spin-charge hybrid qubits, also potentially based on longitudinal interaction through singlet-triplet splitting modulation [1].

Minor points

3.3.5

What is the magnitude of the magnetic field in the double quantum dot generated by the coil?

We reply: To keep the resonator between 4 and 6 GHz, we apply an out-of-plane magnetic field of about 50 - 70 μ T. We have now added a sentence in our manuscript to mention the magnetic field by the coil as follows:

Page 4: To apply a finite magnetic flux, we place a superconducting coil on top of the device which generates an out-of-plane magnetic field of 50 ~ 70 μ T (See Supplementary Note 1).

3.3.6

What is the admixture of light-hole in the characterized Wigner molecules?

We reply: Regarding the light-hole admixture of the WM in Fig. 5, the linewidth of the observed Wigner molecule polariton state (red line in Fig. 5e) is smaller than the one of a bare resonator. This suggests that the contribution from WM state dominates the observed polariton state. Quantitatively, from the effective photon-WM coupling strength, and the measured photon-WM energy detuning we estimate the hybridized polariton state to have around ~ 0.9 (0.1) contribution from the WM (photon) state which indeed depends on the energy detuning.

3.3.7

Please label charge states in the figures (as done in Fig. S5).

We reply: We deeply appreciate the referee's detailed comment. However, being in a multi hole regime as discussed above in 3.3.1, we respectfully decide not to add the charge numbers in the figures because further information, such as the hole number parity, cannot be inferred from the stability diagrams other than the one presented in Supplementary Fig. 5.

References

- [1] J. C. Abadillo-Uriel, Evelyn King, S. N. Coppersmith, and Mark Friesen. Long-range two-hybrid-qubit gates mediated by a microwave cavity with red sidebands. *Physical Review A*, 104(3):032612, September 2021. doi: 10.1103/PhysRevA.104.032612.
- [2] José C. Abadillo-Uriel, Biel Martinez, Michele Filippone, and Yann-Michel Niquet. Two-body Wigner molecularization in asymmetric quantum dot spin qubits. *Phys. Rev. B*, 104(19):195305, November 2021. doi: 10.1103/PhysRevB.104.195305.
- [3] J. Corrigan, J. P. Dodson, H. Ekmel Ercan, J. C. Abadillo-Uriel, Brandur Thorgrimsson, T. J. Knapp, Nathan Holman, Thomas McJunkin, Samuel F. Neyens, E. R. MacQuarrie, Ryan H. Foote, L. F. Edge, Mark Friesen, S. N. Coppersmith, and M. A. Eriksson. Coherent Control and Spectroscopy of a Semiconductor Quantum Dot Wigner Molecule. *Phys. Rev. Lett.*, 127(12):127701, September 2021. doi: 10.1103/PhysRevLett.127.127701.
- [4] Simon Geyer, Bence Hetényi, Stefano Bosco, Leon C. Camenzind, Rafael S. Eggli, Andreas Fuhrer, Daniel Loss, Richard J. Warburton, Dominik M. Zumbühl, and Andreas V. Kuhlmann. Two-qubit logic with anisotropic exchange in a fin field-effect transistor. December 2022.
- [5] Vitaly N. Golovach, Alexander Khaetskii, and Daniel Loss. Spin relaxation at the singlet-triplet crossing in a quantum dot. *Physical Review B*, 77(4):045328, January 2008. doi: 10.1103/PhysRevB.77.045328.
- [6] Wonjin Jang, Min-Kyun Cho, Hyeongyu Jang, Jehyun Kim, Jaemin Park, Gyeonghun Kim, Byoungwoo Kang, Hwanchul Jung, Vladimir Umansky, and Dohun Kim. Single-Shot Readout of a Driven Hybrid Qubit in a GaAs Double Quantum Dot. *Nano Lett.*, 21(12):4999–5005, June 2021. ISSN 1530-6984. doi: 10.1021/acs.nanolett.1c00783.
- [7] Wonjin Jang, Jehyun Kim, Jaemin Park, Gyeonghun Kim, Min-Kyun Cho, Hyeongyu Jang, Sangwoo Sim, Byoungwoo Kang, Hwanchul Jung, Vladimir Umansky, and Dohun Kim. Wigner-molecularization-enabled dynamic nuclear polarization. *Nat Commun*, 14(1):2948, May 2023. ISSN 2041-1723. doi: 10.1038/s41467-023-38649-5.
- [8] Daniel Jirovec, Andrea Hofmann, Andrea Ballabio, Philipp M. Mutter, Giulio Tavani, Marc Botifoll, Alessandro Crippa, Josip Kukucka, Oliver Sagi, Frederico Martins, Jaime Saez-Mollejo, Ivan Prieto, Maksim Borovkov, Jordi Arbiol, Daniel Chrastina, Giovanni Isella, and Georgios Katsaros. A singlet-triplet hole spin qubit in planar Ge. *Nat. Mater.*, 20(8):1106–1112, August 2021. ISSN 1476-4660. doi: 10.1038/s41563-021-01022-2.
- [9] A. C. Johnson, J. R. Petta, C. M. Marcus, M. P. Hanson, and A. C. Gossard. Singlet-triplet spin blockade and charge sensing in a few-electron double quantum dot. *Physical Review B*, 72(16):165308, October 2005. doi: 10.1103/PhysRevB.72.165308.
- [10] Yuan Kang, Zong-Hu Li, Zhen-Zhen Kong, Fang-Ge Li, Tian-Yue Hao, Ze-Cheng Wei, Song-Yan Deng, Bao-Chuan Wang, Hai-Ou Li, Gui-Lei Wang, Guang-Can Guo, Gang Cao, and Guo-Ping Guo. Coupling of hole double quantum dot in planar germanium to a microwave cavity, October 2023.
- [11] Yan Li, Shu-Xiao Li, Fei Gao, Hai-Ou Li, Gang Xu, Ke Wang, Di Liu, Gang Cao, Ming Xiao, Ting Wang, Jian-Jun Zhang, Guang-Can Guo, and Guo-Ping Guo. Coupling a Germanium Hut Wire Hole Quantum Dot to a Superconducting Microwave Resonator. *Nano Letters*, 18(3):2091–2097, March 2018. ISSN 1530-6984. doi: 10.1021/acs.nanolett.8b00272.
- [12] X. Mi, J. V. Cady, D. M. Zajac, P. W. Deelman, and J. R. Petta. Strong coupling of a single electron in silicon to a microwave photon. *Science*, 355(6321):156–158, January 2017. doi: 10.1126/science.aal2469.
- [13] X. Mi, M. Benito, S. Putz, D. M. Zajac, J. M. Taylor, Guido Burkard, and J. R. Petta. A coherent spin–photon interface in silicon. *Nature*, 555(7698):599–603, March 2018. ISSN 1476-4687. doi: 10.1038/nature25769.
- [14] Clemens Müller, Jared H Cole, and Jürgen Lisenfeld. Towards understanding two-level-systems in amorphous solids: insights from quantum circuits. *Reports on Progress in Physics*, 82(12):124501, oct 2019. doi: 10.1088/1361-6633/ab3a7e. URL <https://dx.doi.org/10.1088/1361-6633/ab3a7e>.
- [15] E. Paladino, Y. M. Galperin, G. Falci, and B. L. Altshuler. $\mathbb{1}$ noise: Implications for solid-state quantum information. *Rev. Mod. Phys.*, 86(2):361–418, April 2014. doi: 10.1103/RevModPhys.86.361.

- [16] S. Pecker, F. Kuemmeth, A. Secchi, M. Rontani, D. C. Ralph, P. L. McEuen, and S. Ilani. Observation and spectroscopy of a two-electron Wigner molecule in an ultraclean carbon nanotube. *Nature Phys*, 9(9): 576–581, September 2013. ISSN 1745-2481. doi: 10.1038/nphys2692.
- [17] N. Samkharadze, G. Zheng, N. Kalhor, D. Brousse, A. Sammak, U. C. Mendes, A. Blais, G. Scappucci, and L. M. K. Vandersypen. Strong spin-photon coupling in silicon. *Science*, 359(6380):1123–1127, March 2018. doi: 10.1126/science.aar4054.
- [18] P. Scarlino, D. J. van Woerkom, A. Stockklauser, J. V. Koski, M. C. Collodo, S. Gasparinetti, C. Reichl, W. Wegscheider, T. Ihn, K. Ensslin, and A. Wallraff. All-Microwave Control and Dispersive Readout of Gate-Defined Quantum Dot Qubits in Circuit Quantum Electrodynamics. *Physical Review Letters*, 122(20):206802, May 2019. doi: 10.1103/PhysRevLett.122.206802.
- [19] P. Scarlino, J. H. Ungerer, D. J. van Woerkom, M. Mancini, P. Stano, C. Müller, A. J. Landig, J. V. Koski, C. Reichl, W. Wegscheider, T. Ihn, K. Ensslin, and A. Wallraff. In situ Tuning of the Electric-Dipole Strength of a Double-Dot Charge Qubit: Charge-Noise Protection and Ultrastrong Coupling. *Phys. Rev. X*, 12(3):031004, July 2022. doi: 10.1103/PhysRevX.12.031004.
- [20] I. Shapir, A. Hamo, S. Pecker, C. P. Moca, O. Legeza, G. Zarand, and S. Ilani. Imaging the electronic wigner crystal in one dimension. *Science*, 364(6443):870–875, 2019. doi: 10.1126/science.aat0905. URL <https://www.science.org/doi/abs/10.1126/science.aat0905>.
- [21] Zhan Shi, C. B. Simmons, J. R. Prance, John King Gamble, Teck Seng Koh, Yun-Pil Shim, Xuedong Hu, D. E. Savage, M. G. Lagally, M. A. Eriksson, Mark Friesen, and S. N. Coppersmith. Fast Hybrid Silicon Double-Quantum-Dot Qubit. *Phys. Rev. Lett.*, 108(14):140503, April 2012. ISSN 0031-9007, 1079-7114. doi: 10.1103/PhysRevLett.108.140503.
- [22] Zhan Shi, C. B. Simmons, Daniel R. Ward, J. R. Prance, Xian Wu, Teck Seng Koh, John King Gamble, D. E. Savage, M. G. Lagally, Mark Friesen, S. N. Coppersmith, and M. A. Eriksson. Fast coherent manipulation of three-electron states in a double quantum dot. *Nat Commun*, 5(1):3020, January 2014. ISSN 2041-1723. doi: 10.1038/ncomms4020.
- [23] A. Stockklauser, P. Scarlino, J. V. Koski, S. Gasparinetti, C. K. Andersen, C. Reichl, W. Wegscheider, T. Ihn, K. Ensslin, and A. Wallraff. Strong Coupling Cavity QED with Gate-Defined Double Quantum Dots Enabled by a High Impedance Resonator. *Phys. Rev. X*, 7(1):011030, March 2017. doi: 10.1103/PhysRevX.7.011030.
- [24] Brandur Thorgrimsson, Dohun Kim, Yuan-Chi Yang, L. W. Smith, C. B. Simmons, Daniel R. Ward, Ryan H. Foote, J. Corrigan, D. E. Savage, M. G. Lagally, Mark Friesen, S. N. Coppersmith, and M. A. Eriksson. Extending the coherence of a quantum dot hybrid qubit. *npj Quantum Inf*, 3(1):1–4, August 2017. ISSN 2056-6387. doi: 10.1038/s41534-017-0034-2.
- [25] Jann H. Ungerer, Alessia Pally, Artem Kononov, Sebastian Lehmann, Joost Ridderbos, Patrick P. Potts, Claes Thelander, Kimberly A. Dick, Ville F. Maisi, Pasquale Scarlino, Andreas Baumgartner, and Christian Schönenberger. Strong coupling between a microwave photon and a singlet-triplet qubit. 2023.
- [26] W. G. van der Wiel, S. De Franceschi, J. M. Elzerman, T. Fujisawa, S. Tarucha, and L. P. Kouwenhoven. Electron transport through double quantum dots. *Rev. Mod. Phys.*, 75:1–22, Dec 2002. doi: 10.1103/RevModPhys.75.1. URL <https://link.aps.org/doi/10.1103/RevModPhys.75.1>.
- [27] Rui Wang, Russell S. Deacon, Jian Sun, Jun Yao, Charles M. Lieber, and Koji Ishibashi. Gate Tunable Hole Charge Qubit Formed in a Ge/Si Nanowire Double Quantum Dot Coupled to Microwave Photons. *Nano Letters*, 19(2):1052–1060, February 2019. ISSN 1530-6984. doi: 10.1021/acs.nanolett.8b04343.
- [28] Constantine Yannouleas and Uzi Landman. Molecular formations and spectra due to electron correlations in three-electron hybrid double-well qubits. *Phys. Rev. B*, 105(20):205302, May 2022. doi: 10.1103/PhysRevB.105.205302.
- [29] Cécile X. Yu, Simon Zihlmann, José C. Abadillo-Uriel, Vincent P. Michal, Nils Rambal, Heimanu Niebojewski, Thomas Bedecarrats, Maud Vinet, Etienne Dumur, Michele Filippone, Benoit Bertrand, Silvano De Franceschi, Yann-Michel Niquet, and Romain Maurand. Strong coupling between a photon and a hole spin in silicon. 18(7):741–746, July 2023. doi: 10.1038/s41565-023-01332-3.

Response to reviewers

We thank the referees for their comments and suggestions, to which detailed responses follow below. Changes to the manuscript are highlighted in red in this document.

1 Reviewer #1 (Remarks to the Author)

1.1

The authors have addressed all points raised by the reviewers and made changes to the manuscript and supplementary material wherever necessary. The manuscript has significantly improved in readability and is now much easier to follow and understand. I especially appreciate the added information on additional states interacting with the resonator that have not been discussed before (e.g. two-level systems in the SQUID-array). Therefore, the manuscript in its current form is sound and worth publishing.

We reply: We thank the referee for appreciating our efforts in revising the manuscript and for pointing out that it is now scientifically and formally ready for publication.

1.2

To conclude, the overall message of the manuscript can be summarized as follows:

- 1) first demonstration of strong charge – photon coupling for hole carriers in planar Ge structures
- 2) investigation of low-lying energy states in multi-charge quantum dots by resonant spectroscopy taking advantage of a frequency tunable resonator

I fully acknowledge the achievement of point 1) as an important demonstration for the quantum dot and spin qubit community as planar germanium is a very promising platform for future experiments.

We reply: We thank the referee for pointing this out. Indeed, achieving strong coupling with the charge degree of freedom represents an important first milestone towards spin-photon coupling. It is worth highlighting that, so far, only two other published works have directly demonstrated strong coupling with the charge degrees of electrons (not holes).

The investigation of low-lying energy states by resonant spectroscopy is on its own an interesting technique. However, the presence of such states in multi-charge quantum dots is not at all unexpected (see for example previous reports on spin-charge hybrid qubits). Furthermore, it is obvious as well, that some of these transitions can be more coherent (depending on their spin symmetries). Overall, the information on the low-lying energy states the authors extract from their measurements remain limited. Two-tone spectroscopy and magnetic field dependent measurements would clearly shed more light on the physics involved. Therefore, the novelty of the presented work is limited and a more specialized journal might be more appropriate.

We reply: We respectfully but strongly disagree with the referee. We argue that the explicit demonstration and analysis of such phenomena are not at all obvious. While it may be easy to claim that a phenomenon is predictable, we firmly believe that the significance of a work does not depend on whether the observation is completely unanticipated or not. We would like to emphasize once again that our work systematically elucidates the extraordinary spectra arising from non-trivial spin structures. This provides valuable insights for investigating exchange-coupled spins within a circuit quantum electrodynamics framework.

Planar Ge currently represents the main platform for scaling up spin qubits in quantum processors, and the physics of holes confined in QDs is far from being completely understood. There are still many aspects to study and clarify, especially concerning multiple holes confined in QDs. Therefore, our study is both interesting from a fundamental and technological perspective.

In particular, the observation of transitions involving the spin degree of freedom at zero magnetic field is neither obvious nor trivial. These transitions constitute a very interesting platform for encoding spin-charge hybrid qubits, which can be easily coupled to microwave resonators, as we directly demonstrate in our work.

The fact that similar hybrid states have been demonstrated in silicon (using low-energy valley states) or in GaAs (limited by hyperfine interaction) does not undermine the novelty and importance of studying these states for holes in planar Ge.

1.3

Minor comments:

- The authors implemented a detuning dependent decoherence rate of the charge qubit for the fitting ($\gamma = \gamma_0 + \gamma_\epsilon * \frac{\epsilon/\hbar}{\omega_q}$). They do state the value for γ_0 . However, I was unable to find a value for γ_ϵ , which would give the reader an idea on the magnitude of the charge noise in these structures. Could the authors give an order of magnitude for this parameter? Does it contribute significantly to decoherence or not?

We reply: We thank the referee for the comment. Indeed, the value of Γ_ϵ was reported only in the caption of Supplementary Table 2. Now, its value $\Gamma_\epsilon \sim 164$ MHz is reported also in the Methods section, together with the one of Γ_0 .

Page 15: **We extract a value of $\Gamma_\epsilon \sim 164$ MHz from the fit.**

1.4

- Fig2 (b) misses a label for the y-axis (I guess it should be V_{PR})

We reply: We thank the referee for the accurate observation. We now added the missing label V_{PR} .

2 Reviewer #2 (Remarks to the Author)

2.1

Unfortunately, the revisions made have resulted in a degradation of the theoretical treatment in the manuscript. This is centered around the replacement of the term "Wigner molecule" (which is well established and widely used in the literature for the last 25 years) by the term "Strongly Correlated States (SCSs)" [which is indeterminate, e.g., the Laughlin states in the lowest Landau level (high B) and Kondo-effect states are strongly correlated states].

We reply: We would like to thank the referee for the in-depth comment about the second part of our work, specifically regarding the observation of low-lying energy states via the strongly coupled resonator. The referee identifies these states as Wigner molecular (WM) states, as we also clearly stated in the first version of our manuscript and we are still currently proposing as the most plausible interpretation.

We believe that these WM states constitute a valuable resource for encoding information in QDs and for hybrid superconducting-semiconducting devices. They offer magnetic-field-free operation capability, which is particularly beneficial for interfacing with superconducting circuits. Additionally, WM states have potential applications in longitudinal hole-photon coupling schemes, which could be intriguing in the context of cavity quantum electrodynamics with quantum dots. Furthermore, understanding the physics of multiple holes confined in QDs is crucial from both fundamental and technological perspectives.

We would like to remind the referee that Referee 2 was already very satisfied with the first version of the manuscript and "gladly recommended acceptance." However, we would like to emphasize that the changes implemented during the first round of review were determined by the observations raised by Referee 1, who was not convinced about the Wigner molecularization process based on our analysis and modeling of the reported QD spectroscopy. In light of this, we have revised our terminology in the manuscript to refer to these states as "quenched orbital states due to a strong Coulomb correlation" or "strongly correlated states" (SCS) rather than "Wigner Molecule (WM) states", and modified the manuscript accordingly.

Even more importantly, as Referee 2 is now suggesting, several theoretical works have analyzed the phenomena of WM in multi-particle QDs. We have added some of the suggested references in the current version of the manuscript.

To further stress the important role played by the WM process, we have also modified the introduction of our manuscript as follows:

Page 2: When further enhanced by anisotropic QD confinement [39], these states can lead to excitation energies below 10 GHz that have been observed in GaAs [40, 41], Si [42], and carbon nanotube [43, 44] QDs and attributed to Wigner molecular states [38, 45-50].

Moreover, please note that we are still recognizing Wigner molecularization as the most plausible explanation for our observation as reported below, where we have now included more references to further support this claim:

Page 10: Although a more comprehensive investigation based on full-configuration-interaction calculations is necessary to precisely characterize the energy scales within the DQD [42, 46], our preliminary analysis provides evidences that the observed features in Fig. 5a may be attributed to WM states [38, 45-50].

Page 12: While a detailed measurement of the charge density distribution of the ground state is required to unambiguously prove the Wigner molecularization process [44], the presence of strong Coulomb correlation and QD confinement anisotropy, as suggested by our simulations, make Wigner molecules the most plausible model to describe the observed quenching of strongly correlated states in our QDs [38, 45-50].

We would like to emphasize our willingness to restore the original title and further highlight the role of the WM process if the appeal is successful and the manuscript will be published in Nature Communications. We also wish to reiterate that the substance of our work and manuscript has not changed since our first revision. Whether we refer to these states as WM states or use another term (we welcome any suggestions from the referees), it does not change their presence, the significance of studying them, or the fact that we have implemented their first study using a strongly coupled frequency-tunable resonator.

Furthermore, we would like strongly to remark that the ongoing debate between the suggestions from Referee 1 and Referee 2 regarding the nature of these states indicates a healthy debate in the field, further supporting the publication of our work.

2.2

The following points elaborate further and provide clarifications of the statement above:

1) The manuscript attributes the appearance of the SCSs in the planar Ge DQD (at zero or low magnetic field, B) to larger than unity values of the Wigner ratio λ_W , which expresses the ratio of the Coulomb potential energy over the kinetic energy.

Independently of the number of electrons (N), large values of λ_W induce electron localization (Wigner conceived the “Wigner crystal” of electrons under the condition that the potential energy dominates the kinetic energy; see [R1] Phys. Rev. 46, 1002 (1934), <https://journals.aps.org/pr/abstract/10.1103/PhysRev.46.1002>) For a small number $N \leq 10$ of confined 2D or quasi-1D electrons, full-configuration- interaction (FCI) and Quantum Monte-Carlo calculations for large λ_W have confirmed the electron localization and formation of Wigner molecules (WMs), finite-size analogs of the Wigner crystal). Electron localization and WM formation for larger N’s (up to 30) have been also found by employing unrestricted Hartree-Fock calculations (see, e.g., [R2] PRL 82, 5325 (1999) <https://journals.aps.org/prl/pdf/10.1103/PhysRevLett.82.5325>, and [R3] arXiv:2406.05886, <https://arxiv.org/abs/2406.05886> (for a case with strong deformation)).

We reply: We agree with the referee on this point and we are grateful for the suggested references. We explicitly added the reference [R2].

2.3

2) The fact that heavy holes form WMs for $\lambda_W > 1$ has been recently confirmed experimentally via STM imaging of the 2D charge densities in bilayer TMD moiré superlattices; see [R4] Hongyuan Li et al., arXiv:2312.07607, <https://arxiv.org/abs/2312.07607>

We reply: We thank the referee to point this out. We were not aware about this work showing an STM imaging of the WM states in a hole 2D system.

2.4

3) The fact that WM formation is accompanied by a strong suppression of energy gaps in the excitation spectra is well documented in the literature. For N=2-5 1D electrons, see the converging (as a function of size) energy spectra in [R5] Phys. Rev. B 70, 035401 (2004), <https://journals.aps.org/prb/abstract/10.1103/PhysRevB.70.035401> For N=4 electrons in a 2D double QD, see [R6] Phys. Rev. B 80, 045326 (2009), <https://journals.aps.org/prb/abstract/10.1103/PhysRevB.80.045326> See also [R7] Phys. Rev. B 104, 235302 (2021) <https://journals.aps.org/prb/abstract/10.1103/PhysRevB.104.235302>, and Refs. [38-45] in the revised manuscript.

We reply: We thank the referee for pointing out these theoretical works on multiple particles confined in QDs. Indeed, the mentioned works provide important theoretical support for the interpretations of our spectroscopy measurements in the context of WM states. We have now added references [R5] and [R6] to the manuscript.

2.5

4) Given the current stage of discussion/debate that arose in the context of the refereeing of this manuscript, as written, the short section 6 in the supplementary notes is inadequate. Namely, (i) The title is confusing. I do not understand the sentence “Orbital state renormalization due to strong correlation”. The word “orbital” usually refers to a single-determinant wave function and to a mean-field approach. However, here an exact 2e wave function is used, which cannot be represented by a single determinant alone, a property shared with the FCI wave functions, and in this context “orbital state” is not appropriate. I suggest that the original title of the first submission be restored.

We reply:

We appreciate the referee’s comments and suggestions regarding section 6 in the supplementary notes. We understand the confusion surrounding the title “Orbital state renormalization due to strong correlation.” Our intention in this section was to demonstrate the effect of Coulomb interactions, showing how they transition from negligible to significant. For this analysis, we compare the changes in energy relative to the orbital energy, derived from the single particle picture, as Coulomb interactions are included.

However, we acknowledge the potential for misunderstanding and thank the referee for highlighting this issue. We have revised the title of this section to “Coulomb interactions in doubly occupied QDs and WM formations” to better reflect the content and include the WM process in the title.

2.6

(ii) The fact that the WM is a well-established concept is not conveyed in this section (and thus in the manuscript). I suggest that the single Ref. [18] in the sentence “... which is often referred to as a Wigner molecule [18]” in this section be expanded to a full list reflecting the earlier abundant WM literature, including Refs. [R1-R7] above, Refs. [38-45] in the manuscript, and the additional references below.

We reply: We understand the point raised by the referee and we have now added some of the new references suggested by the referee.

2.7

Recommendation: The paper represents a solid advancement to the experimental effort toward the understanding and control of planar Ge-based qubits. As is, however, it is not ready for publication. I suggest that major revisions are needed to convey the proper synergy between the Wigner ratio, QD anisotropy, and formation of Wigner molecules/finite Wigner crystallites.

SOME ADDITIONAL WIGNER-MOLECULE LITERATURE

- [R8] Eur. Phys. J. D 16, 373-380 (2001) <https://epjd.epj.org/articles/epjd/abs/2001/10/iss-89/iss-89.html?mb=0>
- [R9] Phys. Rev. B 65, 075309 (2002) <https://journals.aps.org/prb/abstract/10.1103/PhysRevB.65.075309>
- [R10] Phys. Rev. B 65, 115312 (2002) <https://journals.aps.org/prb/abstract/10.1103/PhysRevB.65.115312>
- [R11] Phys. Rev. B 67, 045311 (2003) <https://journals.aps.org/prb/abstract/10.1103/PhysRevB.67.045311>
- [R12] Nature Physics 2, 336-340 (2006) <https://www.nature.com/articles/nphys293>
- [R13] Phys. Rev. Lett. 85, 1726 (2000) <https://journals.aps.org/prl/abstract/10.1103/PhysRevLett.85.1726>
- [R14] Phys. Rev. B 77, 035325 (2008) <https://journals.aps.org/prb/abstract/10.1103/PhysRevB.77.035325>
- [R15] J. Chem. Phys. 124, 124102 (2006) <https://doi.org/10.1063/1.2179418>

We reply:

We thank the referee for reiterating in this second review round that our work represents a solid advancement in the experimental effort toward the understanding and control of planar Ge-based qubits.

In this current revision, we have clearly emphasized the significant role played by the Wigner Molecularization (WM) process in the observations reported in the second part of our manuscript. As mentioned previously, we are willing to restore the original title and further highlight the role of the WM process if the appeal is successful and the manuscript will be published in Nature Communications.

3 Reviewer #3 (Remarks to the Author)

3.1

The authors answered well to my points and changed the manuscript accordingly. The rearrangements of figure panels improved the readability.

We reply: We thank the referee for acknowledging the improvement of our manuscript.

About significance of the work, the authors argue that "planar Ge is currently at the forefront of spin qubit technology, as evidenced by the substantial volume of recent literature utilizing Ge-based hole QDs." I fully agree that planar Ge is one of the promising qubit platforms.

We reply: Indeed, there is significant current interest and rapid development in spin quantum processors in planar Ge, as evidenced by the very recent publication in Science [C.A. Wang *et al.*, *Science* **385**, 6707 (2024)]. Additionally, a new paper appeared on the Arxiv discussing the strong coupling of holes in planar Ge to a grAl resonator [M. Janik *et al.*, *arXiv:2407.03079* (2024)]. Our work represents the first direct demonstration of strong hole-photon coupling.

In other qubit platforms such as holes in MOS devices or electrons in planar Si quantum wells, strong coupling and even coherent spin-coupling was demonstrated. Therefore, doubts remain whether the presented progress justifies publication in Nature Communications.

We reply: We would like to strongly stress that our work does not just represent the first direct strong charge-photon coupling for holes in the currently most promising platform for spin qubits quantum processors, but it goes beyond by making use of the strong coupling to study in-depth the unanticipated spectra related to strongly-correlated states in hole-based QDs using a frequency tunable resonator. Indeed, the physics of holes in semiconducting QDs is still largely unexplored and the presented hybrid platform can definitively represent a very useful tool in this regards.

The fact that both strong spin-photon coupling and also spin-spin coupling have been demonstrated in some platforms (mainly for electrons), does not imply that it is trivial and obvious to reproduce in novel platforms. Indeed, on planar Ge different groups have been studying the compatibility with superconducting resonators without achieving it.

Additionally, it is worth highlighting that so far only two other published works have demonstrated strong coupling with charge degrees of electrons (not holes). Our work represents the first direct demonstration of strong coupling with the charge degree of holes, making our work unique in its achievements. This result is important not only for QD scalability, but also for achieving in the future high-fidelity and fast readout.

Given the significance, our work is expected to gain attention from the entire community working on QDs and thereby we believe our work suits the aim and scope of Nature Communications.